# Resolution dependence of magnetosheath waves in global hybrid-Vlasov simulations

Maxime Dubart[1], Urs Ganse[1], Adnane Osmane[1], Andreas Johlander[1], Markus Battarbee[1], Maxime Grandin[1], Yann Pfau-Kempf[1], Lucile Turc[1], and Minna Palmroth[1,2]

[1]Department of Physics, University of Helsinki, Helsinki, Finland
[2]Space and Earth Observation Centre, Finnish Meteorological Institute, Helsinki, Finland

**Correspondence:** Maxime Dubart (maxime.dubart@helsinki.fi)

**Abstract.** Kinetically driven plasma waves are fundamental for a description of the thermodynamical properties of the Earth's magnetosheath. The most commonly observed ion-scale instabilities are generated by temperature anisotropy of the ions, such as the mirror and proton cyclotron instabilities. We investigate here the spatial resolution dependence of the mirror and proton cyclotron instabilities in a global hybrid-Vlasov simulation using the Vlasiator model, in order to find optimal resolutions and help future global hybrid-Vlasov simulations to save resources when investigating those instabilities in the magnetosheath. We compare the proton velocity distribution functions, power spectra and growth rates of the instabilities in a set of simulations with three different spatial resolutions but otherwise identical set-up. We find that the proton cyclotron instability is absent at the lowest resolution and that only the mirror instability remains, which leads to an increased temperature anisotropy in the simulation. We conclude that the proton cyclotron instability, its saturation and the reduction of the anisotropy to marginal levels are resolved at the highest spatial resolution. A further increase in resolution does not lead to a better description of the instability to an extend that would justify this increase at the cost of numerical resources in future simulations. We also find that spatial resolutions between 1.32 and 2.64 times the inertial length in the solar wind present acceptable limits for the resolution within which the velocity distribution functions resulting from the proton cyclotron instability are still bi-maxwellian and reach marginal stability levels. Our results allow us to determine a range of spatial resolution suitable for the modelling of the proton cyclotron and mirror instabilities and should be taken into consideration regarding the optimal grid spacing for the modelling of these two instabilities, within available computational resources.

## 1 Introduction

The Earth's magnetosheath is permeated with several kinds of ion-kinetic waves, which are an important source of energy transfer and dissipation within the magnetosheath plasma (Schwartz et al., 1997). The most commonly observed waves arise from instabilities generated by temperature anisotropy of the ions. The mirror instability (Chandrasekhar et al., 1958; Hasegawa, 1969; Southwood and Kivelson, 1993; Kivelson and Southwood, 1996) and the proton cyclotron instability (Kennel and Petschek, 1966; Davidson and Ogden, 1975; Gary et al., 1993) are excited by a temperature anisotropy where the ions' perpendicular temperature $T_\perp$ is larger than the parallel temperature $T_\parallel$.

The mirror instability gives rise to compressional, linearly polarised waves characterised by zero frequency in the plasma frame, anti-correlation between the plasma density and the magnetic field, and magnetic perturbations which are mostly parallel to the background magnetic field (Tsurutani et al., 1982; Price et al., 1986). They create magnetic mirror-like structures trapping particles (Soucek et al., 2008). The proton cyclotron instability has maximum growth rate around the ion cyclotron frequency, and produces waves propagating in the direction parallel to the background magnetic field. The magnetic perturbations of the waves are perpendicular to the background magnetic field and produce left-handed nearly circularly polarised waves in the plasma frame (Davidson and Ogden, 1975; Lacombe et al., 1994; Remya et al., 2014). Both instabilities isotropize the ion populations of the magnetosheath by pitch angle scattering the protons (Hasegawa, 1969; Tanaka, 1985), thus reducing the temperature anisotropy of the population. The proton cyclotron instability isotropizes ions faster than the mirror instability (McKean et al., 1992).

The mirror and proton cyclotron instabilities have been observed in the Earth's magnetosheath (Tsurutani et al., 1982; Anderson et al., 1996; Gary et al., 1993; Soucek et al., 2008) as well as in the solar wind (Hellinger et al., 2006), heliosheath (Liu et al., 2007; Tsurutani et al., 2011), and sheath regions driven by coronal mass ejections (Ala-Lahti et al., 2019). The mirror instability has also been observed in the magnetosheaths of Jupiter and Saturn (Tsurutani et al., 1982; Erdős and Balogh, 1996), distant magnetotails (Tsurutani et al., 1984), cometary sheaths (Russell et al., 1987; Glassmeier et al., 1993; Tsurutani et al., 1999) and interplanetary space (Tsurutani et al., 1992).

The competition between the mirror and proton cyclotron instabilities has been a topic of many studies for the past years (Price et al., 1986; Winske and Quest, 1988; Brinca and Tsurutani, 1989; Gary, 1992; Gary et al., 1993; Gary and Winske, 1993; Anderson and Fuselier, 1993; Lacombe and Belmont, 1995; Schwartz et al., 1997; Shoji et al., 2009, 2012; Remya et al., 2013). The proton cyclotron instability is dominant in lower beta plasma (Gary, 1992), while mirror modes are dominant in high plasma beta (Tsurutani et al., 1982). Although the Earth's magnetosheath tends to have a high plasma beta, there are also many reports of observations of proton cyclotron waves (Remya et al., 2014; Soucek et al., 2015; Zhao et al., 2018, 2020). Soucek et al. (2015) show observations in the Earth's magnetosheath of proton cyclotron waves associated with Alfvén Mach number below 7, which is within the typical Mach number range at Earth, between 6 and 8 (Winterhalter and Kivelson, 1988). Price et al. (1986) showed that the presence of $He^{++}$ tends to lower the growth rate of the proton cyclotron instability, while Brinca and Tsurutani (1989) showed that such ions would have less effect on the growth rate of the mirror instability. The mechanisms of the competition between the two instabilities have also been extensively studied using numerical simulations of the magnetosheath plasmas (Shoji et al., 2009, 2012; Remya et al., 2013). However, in this study, instead of investigating which mode is dominant, we rather evaluate the influence of the spatial resolution of a simulation in the development of those instabilities. The goal of this study is to determine an acceptable resolution where both instabilities are resolved correctly.

The properties of both instabilities have been studied through simulations by, e.g. McKean et al. (1994); Gary and Winske (1993); Seough et al. (2014); Hoilijoki et al. (2016). However, these simulations were either one-dimensional (Gary and Winske, 1993) or used a particle-in-cell approach (Seough et al., 2014), and apart from Hoilijoki et al. (2016), none of them studied the instabilities in a global simulation of Earth's magnetosheath. Gary and Winske (1993) and Remya et al. (2013) showed that the introduction of Helium ions reduces the growth rate of the proton cyclotron instability. Seough et al. (2014)

found good agreement between quasi-linear theory and the simulation. Hoilijoki et al. (2016) presented the first study of mirror modes in a global hybrid-Vlasov simulation and found that their properties were consistent with that obtained from previous studies. Kunz et al. (2014) also showed in a hybrid-kinetic simulation that trapped particles govern the nonlinear evolution of the mirror instability to maintain the pressure anisotropy to a marginal level. They also found that energy is removed below the Larmor scales by what appears to be Kinetic Alfvén Waves turbulence (Sahraoui et al., 2006; Howes et al., 2011). These simulations did not study the impact of the spatial resolution on the description of the instabilities.

Modern plasma physics is increasingly relying on the support of numerical simulations in understanding waves and instabilities. Whether it is used for the study of laboratory plasmas (Revel et al., 2018), nuclear fusion (Görler et al., 2011) or space plasmas (McKean et al., 1994), numerical modelling of instabilities is crucial for the understanding of the physics of the system. However, no matter what kind of numerical model is chosen, it is difficult to model the entire system at a numerical resolution capturing both large-scale and small-scale physical processes involved without incurring a very high computational cost. The issue is even more relevant when global simulations of large systems are carried out, for physical understanding or for space weather forecasting (Palmroth et al., 2018; Pomoell and Poedts, 2018). The choice of resolution is a central parameter in numerical models, and often presents a tradeoff between accuracy and computational cost. To be able to make an informed choice about this tradeoff, a firm understanding of the impact of the models' resolution on the physical processes at play in the system is required.

In order to model the processes involved in energy transfer and dissipation, the understanding of the instabilities generating them is essential. In this study, we investigate the impact of the spatial resolution on the ion-scale waves produced by the mirror and proton cyclotron instabilities in a 2D global hybrid-Vlasov simulation of the Earth's magnetosphere using the Vlasiator model (Palmroth et al., 2013; von Alfthan et al., 2014; Palmroth et al., 2018). A study by Pfau-Kempf et al. (2018) showed that, for 1D simulations of oblique shocks, a coarse resolution in Vlasiator such as cells of size $\Delta r = 1000$ km was still sufficient to describe correctly most of the kinetic effects related to shocks. Despite not resolving the ion inertial length in the solar wind of 228 km in this simulation, the results were similar to a simulation with a spatial resolution of $\Delta r = 200$ km, where the ion inertial length was resolved. Ion velocity distribution functions obtained by the Vlasiator model at coarse resolution are also consistent with observations (Kempf et al., 2015). However, the effect of spatial resolution on the description of plasma instabilities in 2D global kinetic simulations is still an open question. The spatial resolution of a simulation impacts the evolution of the fields during the simulation. Therefore, it will modify the development of the different instabilities present in the magnetosheath and their effect on the velocity distribution functions. In this paper, we determine the lowest possible spatial resolution which can still be used to model the mirror and proton cyclotron instabilities in a 2D global simulation. This allows computational resources to be used more efficiently when global hybrid-Vlasov simulations of near-Earth space are expanded to the third dimension. We focus this investigation on the magnetosheath waves downstream of the quasi-perpendicular shock, as they are well defined and less perturbed by the shock processes than downstream of the quasi-parallel shock. We decided to focus on the proton cyclotron and mirror instabilities as their properties are well documented and are a good proxy for their dependence on the resolution.

## 2 Global hybrid-Vlasov model

We performed this study using the Vlasiator model. Vlasiator is a global hybrid-Vlasov model (Palmroth et al., 2013; von Alfthan et al., 2014; Palmroth et al., 2018). Currently, it consists of a cartesian 2D spatial grid containing the nightside and dayside of the Earth's magnetosphere, magnetosheath, bow shock and foreshock. A cartesian 3D velocity space grid is coupled with each of the ordinary space cells. The model solves the time evolution of the protons in phase space by solving the Vlasov equation, coupled with the electric and magnetic fields. The fields are propagated using Maxwell's equations. Closure of the system is performed with the generalised Ohm's law including the Hall term. In each grid cell, the protons are discretised as velocity distribution functions (VDFs). Electrons are considered a cold, massless, charge-neutralising fluid. The Vlasiator model, and global hybrid-Vlasov simulations in general, have the advantage to be noise-free (Palmroth et al., 2018).

The Vlasiator model can be run in 1D, or 2D in ordinary space. In this study we investigate the ion-scale waves produced by the proton cyclotron and mirror instabilities in three 2D simulations with different spatial resolutions but otherwise identical set-up. Typically, in 2D Vlasiator simulations, the spatial resolution of the grid in ordinary space is set to $\Delta r = 300$ km (e.g. Blanco-Cano et al., 2018; Grandin et al., 2019; Hoilijoki et al., 2019), which corresponds to $\Delta r = 1.32$ $d_{i,SW}$, where $d_{i,SW}$ is the ion inertial length in the solar wind. Simulations with resolution of $\Delta r = 228$ km $= 1$ $d_{i,SW}$ have also been used (e.g Palmroth et al., 2018; Turc et al., 2018). When extending simulations to 3D, such a high resolution may become unfeasible, even when using adaptive mesh refinement for regions of interest.

In order to study the effect of spatial resolution on magnetosheath waves, we conducted three simulations using the same set-up, with different spatial resolution: $\Delta r = 300$, 600, and 900 km, which corresponds to $\Delta r = 1.32$, 2.64 and 3.96 $d_{i,SW}$ in the solar wind, respectively. The system is in the geocentric solar ecliptic (GSE) coordinate system, assuming a zero magnetic dipole tilt. All runs are 2D, describing the noon-midnight meridional plane (X-Z) of near-Earth space. The real-space boundaries of the simulations extend from $X = -48$ $R_E$ in the nightside to $X = 64$ $R_E$ in the dayside, and from $Z = -60$ $R_E$ to $Z = 40$ $R_E$ in the north-south direction, asymmetrical to accommodate the foreshock in the negative Z-direction, with $R_E = 6371$ km the Earth radius. The north, south and nightside boundaries all apply Neumann boundary conditions. The inner boundary is located at $4.7$ $R_E$ from the centre of the Earth and consists of a perfectly conducting sphere. The homogeneous and constant solar wind is flowing from the dayside boundary in the $-X$ direction with a velocity of 750 km/s, interplanetary magnetic field (IMF) strength of 5 nT, and temperature of 0.5 MK. The IMF makes an angle of $45°$ with respect to the X direction, southward. The solar wind protons are represented by a Maxwellian distribution function, with density 1 cm$^{-3}$, and a velocity space resolution of 30 km s$^{-1}$. This setup is identical to the one used in Blanco-Cano et al. (2018).

Figure 1 displays a global overview of the magnetic field magnitude in the dayside of near-Earth space in the three different runs. One can identify the upstream solar wind (in dark blue), the bow shock, the magnetosheath and the magnetosphere (mostly yellow). The white circle of radius $4.7$ $R_E$ represents the inner boundary of the simulation. The white square indicates the portion of the simulation we will focus on in this study. One can already notice differences in the magnetosheath wave properties as a function of the resolution of the three different setups. For example, we can observe stripes of roughly constant

magnetic field strength in the 300 km run. These structures are larger in the 600 km run, and have almost disappeared in the 900 km run.

## 3 Results: Ion-scale waves

### 3.1 Alfvén waves and mirror modes

In order to identify the resolved wave modes in the different runs, we use the 2D Fast Fourier Transform (FFT) analysis. Figure 2 displays the wave power of the electric field component in the GSE y direction (i.e. the out-of-plane direction), in the simulation frame, as a function of the frequency $\omega$ normalised to the local ion cyclotron frequency in the magnetosheath $\Omega_c = qB/m_p$, where $m_p$ is the proton mass and $q$ the proton charge, the wave vector $\mathbf{k}$ parallel to the average magnetic field over the time and space intervals in panels (a), (c) and (e), and perpendicular in panels (b), (d) and (f). This analysis

is performed in a square extending from $X = 3$ $R_E$ to $X = 6$ $R_E$ and from $Z = 15$ $R_E$ to $Z = 18$ $R_E$, depicted in red in Fig. 1, during a time interval from 800 s to 1200 s of the simulation. The maximum possible $\mathbf{k}$, the Nyquist wave number, depends on the spatial resolution $\Delta r$ of the simulation as $k_{\mathrm{max}} = \pi/\Delta r$, hence a smaller $k_{\mathrm{max}}$ at lower resolution. The x-axis is normalised to the local ion inertial length in the magnetosheath given by $\mathrm{d_{i,M}} = \sqrt{m_p \epsilon_0 c^2/(nq^2)} \approx 135$ km, where $\epsilon_0$ is the vacuum permittivity, $c$ the speed of light, and $n$ the local proton number density in the magnetosheath. The solid black lines

indicate the Courant-Friedrichs-Lewy (CFL) condition (Courant et al., 1967). The CFL condition is a necessary condition for the convergence of the solution in a model and depends on its spatial and temporal resolutions, implying that no signal can propagate more than one spatial cell within one time interval of the simulation. This means that all features found in between the two black lines are beyond the resolution of the simulation and probably result of numerical features.

At the highest resolution, the dominant wave mode observed in Fig. 2a (i.e. $k_\parallel \mathrm{d_{i,M}} \approx -0.4$, $\omega/\Omega_{ci} \approx 1.0$) matches the

145 Alfvén velocity $v_A = B/\sqrt{\mu_0 \rho_m}$ (Alfvén, 1942), indicated by solid blue lines, where $B$ is the magnetic field, $\mu_0$ the vacuum permeability and $\rho_m$ the mass density of protons. This wave mode is propagating almost entirely in the anti-parallel direction, as evidenced by the much smaller wave power along the Alfvén velocity in the perpendicular direction (Fig. 2b). Since the plasma flow in the magnetosheath is super-Alfvénic, two curves describing the Alfvén velocity appear on the left side of Fig. 2a: the upper solid blue curve describes the waves propagating in the direction anti-parallel to the magnetic field in the

150 plasma frame, while the lower solid blue curve describes the waves propagating in the direction parallel to the magnetic field in the plasma frame. These ones appear on the $k_\parallel < 0$ side of the plot, because of the Doppler shift $\omega' = \omega - \mathbf{k} \cdot \mathbf{V}$, where $\mathbf{V}$ is the plasma bulk velocity. As in observations from Zhao et al. (2020), we see both parallel and anti-parallel propagating waves at the same time. The wave mode in Fig. 2a extends up to the proton cyclotron frequency. Figures 2c and 2d show similar features: the observed waves are the same as in Fig. 2a, matching the Alfvén velocity, except that their excitation seems to be

constrained to frequencies below $\approx 0.7 \, \omega_{ci}$. Now for the lowest resolution case, displayed in Fig. 2e and 2f, it appears that the features present in the two higher-resolution simulations are completely absent. The waves around the cyclotron frequency are not resolved at this resolution.

Figure 3 displays the wavelet power spectra obtained from wavelet analysis (Torrence and Compo, 1998) of the out-of-plane component of the magnetic field during the interval of time when the FFTs were performed, taken at the virtual spacecraft locations indicated by a black and white circle in Fig. 1. The frequency of the main wave mode at the highest resolution in panel (a) fluctuates around the proton cyclotron frequency $f_{ci} = \Omega_c/2\pi$ shown with a black line, as observed in Fig. 2a. In panel (b), the waves are still present around the cyclotron frequency but the wave power is lower, as observed in Fig 2c, most likely due to the lower resolution. They are completely absent at the lowest resolution in panel (c), as only very low frequency waves below the cyclotron frequency can be observed.

The polarisation of the magnetic field taken at the virtual spacecraft locations indicated in Fig. 1 is also analysed using the minimum variance analysis (Sonnerup and Scheible, 1998), and the results are displayed in Fig. 4a-d. These hodograms display the magnetic field fluctuations during 18 s. The wave vector is along the minimum variance direction $\delta B_N$. Figures 4a and 4b highlight that, at $\Delta r = 300$ km, the wave displays few to no perturbations in the parallel direction to the magnetic field (panel (a)), and is left-handedly polarised (panel (b)) in the simulation frame. The angle between the wave vector $\mathbf{k}$, obtained from minimum variance analysis, and the ambient magnetic field is $\theta_{kB} = 15°$. This can be assumed to be a nearly parallel propagation. Based on Fig. 2, we find that these waves move along the plasma flow. The frequency of the waves in the plasma frame is given by $\omega' = \omega - \mathbf{k} \cdot \mathbf{V}$. Therefore, a Doppler shift will not change the sign of $\omega$ and the polarisation is the same in the plasma frame. Figures 4c and 4d highlight an identical behaviour of the wave at lower resolution $\Delta r = 600$ km. We don't perform a minimum variance analysis on the third run at $\Delta r = 900$ km because there is no significant wave activity around the ion cyclotron frequency. However, we display the fluctuations of the magnetic field and the proton density at the position indicated in Fig. 1c for the time interval considered in the study in Fig. 5c. This indicates that they are anti-correlated and would suggest the presence of mirror waves (Hoilijoki et al., 2016). Fig. 5a and Fig. 5b show the fluctuations at the other two resolutions. Fig. 5a highlights the presence of higher frequency waves on top of the lower fluctuations. These higher-frequency waves are greatly attenuated in Fig. 5b.

To further analyse how the spatial resolution impacts the different wave modes, we investigate the growth rates of the waves. We use the numerical dispersion solver HYDROS (HYbrid Dispersion RelatiOn Solver) (Told et al., 2016), designed for hybrid kinetic plasmas. The solver assumes a bi-Maxwellian proton distribution function, and we input the ion parallel temperature, the ion temperature anisotropy, and the ion parallel beta in the magnetosheath of the different simulations, taken at the same locations as the data for the wavelet analysis, indicated in Fig. 1, averaged over the time range of the study. The electrons are modelled as a fluid. The propagation angle between the wave vector and the magnetic field vector is set to zero for parallel propagation. Figure 6a displays the growth rate $\gamma$ of the proton cyclotron instability for the three different resolutions. The theoretical maximum wave vector is $k_{\max} = \pi/\Delta r$. However, a signal modelled by such $k_{\max}$ would be described with only two points per wavelength. Slightly more realistic is the assumption that a wave needs to be modelled with at least four cells per wavelength. Hence, we consider that the minimum wavelength the model can resolve at each resolution is $\lambda_{\min} = 4\Delta r$, which corresponds to a maximum wave vector $k_{\max} = \pi/2\Delta r$. This wave length is displayed by vertical dashed lines. The growth rates are consistent with what is observed in Fig. 2: at $\Delta r = 300$ km, the growth rate for the proton cyclotron instability is almost fully within the resolved wave length domain. At $\Delta r = 600$ km, only the low-wavenumber edge of the growth rate curve is

resolved, below $k_{\parallel}d_{i,M} \approx 0.35$, which is consistent with where the wave power vanishes in Fig. 2c. At $\Delta r = 900$ km, the growth rate curve is completely outside the resolved domain. This would explain why the waves around the proton cyclotron frequency are not observed in this run. We also notice that the maximum growth rate is dependent on the resolution of the simulation, the maximum being higher at lowest resolution, because of higher temperature anisotropy. Figure 6b displays the growth rate of the mirror instability for the three different resolutions. We used the same input parameters as for the proton cyclotron instability, except that we set the propagation angle to $45°$, and frequency to zero. At $\Delta r = 300$ km, the growth rate is fully within the resolved domain, but very low for these conditions. It is higher for the two other resolutions, even though only partially resolved.

Panels (a)-(c) of Fig. 7 display the y-component (out of plane) of the magnetic field in the three different runs considered in the study. Waves with high frequency are present at the highest resolution in panel (a), but are absent at the lowest resolution in panel (c). At this resolution, we have found that the waves with a frequency around the proton cyclotron frequency are absent. Panels (d)-(f) of Fig. 7 displays the temperature anisotropy in the three different runs considered in the study. The anisotropy grows downstream of the quasi-perpendicular bow shock. For $\Delta r = 300$ km in panel (a), waves reduce the anisotropy quickly in the middle of the magnetosheath. We have found that waves with a frequency around the proton cyclotron frequency are present at this resolution. For $\Delta r = 600$ km in panel (b), the anisotropy is reduced but at a much slower rate, resulting in larger anisotropy levels in the magnetosheath. At this resolution, the waves at the proton cyclotron frequency have a lower wave power. In panel (c), at $\Delta r = 900$ km, a strong anisotropy persists for more than 5 $R_E$, and even after some isotropization has taken place due to the mirror instability, a high anisotropy, with $T_\perp/T_\parallel \sim 3$, remains.

## 3.2 Velocity distribution functions

Figure 8 displays the magnitude of the magnetic field in a zoomed portion of the simulation in the magnetosheath, downstream of the quasi-perpendicular shock, indicated by the white square in Fig. 1, with velocity distribution functions (VDF) taken from the point marked by the black and white circle (see Fig. 1), for the three different resolutions. In Fig. 8a, small wavelength waves (of the order of $0.2$ $R_E$) are distinguishable. These are the waves with frequency around the proton cyclotron frequency identified in the previous section. In addition, larger wavelength structures (of the order of 1 $R_E$) are observed, becoming larger in Fig. 8e, and becoming distinct magnetic field enhancements in Fig. 8i. These structures appear to be convected with the plasma flow, as shown by the animated version of Figure 1 (see Supplementary Video), and are consistent with the mirror modes identified in the previous section. On the right, three slices of the VDFs through the velocity space in different planes are presented. All velocities are transformed to the local plasma frame. Panels (b), (f) and (j) display the slice in the $(\mathbf{v_B}, \mathbf{v_{B\times V}})$ plane. Panels (c), (g) and (k) display the $(\mathbf{v_{B\times V}}, \mathbf{v_{B\times(B\times V)}})$ plane. Panels (d), (h) and (l) display the $(\mathbf{v_B}, \mathbf{v_{B\times(B\times V)}})$ plane. On panels (b), (c) and (d), corresponding to the $\Delta r = 300$ km resolution, one can identify nearly Maxwellian VDFs in all three directions, which is consistent with observations (Williams et al., 1988). In panels (f), (g) and (h), at $\Delta r = 600$ km, the VDFs have a nearly Maxwellian shape in all three directions, with the beginning of the development of a small loss cone in the parallel direction. At the lowest resolution $\Delta r = 900$ km, in panel (i), the smaller wavelength structures which can be observed in the background of panel (a) have disappeared, with only large structures remaining. Moreover, the associated VDFs in panels (j)

and (l) have an "hourglass" shape, in contrast to the nearly Maxwellian shape in panels (b), (c) and (d). The 600 km case in panels (e)-(h) can be considered an intermediate case.

## 4    Discussion

In this paper, we use Vlasiator simulations of near-Earth space with three different spatial resolutions to investigate the behaviour of the proton cyclotron and the mirror instabilities and their dependence on these resolutions. We used 2D-FFT and wavelet analysis in order to identify the waves produced by these instabilities, their different properties at the different resolutions, and the impact of these different properties on the velocity distribution functions of the protons. The growth rate of the proton cyclotron instability at the different resolutions is calculated and compared with resolution-dependent minimum

wavelengths. The temperature anisotropy in the magnetosheath of the different run is analysed.

As Fig. 2, 3 and 4 illustrate, the higher frequency waves propagate with the Alfvén velocity, in the parallel direction, with perpendicular perturbations which are left-hand nearly circularly polarised, with frequency around the ion cyclotron frequency. This suggests that the wave mode present at the 300 km and 600 km resolutions is the Alfvén ion cyclotron wave mode (AIC waves) (Anderson et al., 1996; Rakhmanova et al., 2017), or also known as electromagnetic ion cyclotron (EMIC) waves. In

the 900 km case, the AIC waves are not present. In addition, the magnetic field and density perturbation analysis shown in Fig. 5c suggest that mirror modes are present in the 900 km case, which appear to be the dominant wave mode at this spatial resolution. The anti-correlation between density and magnetic field strength is less pronounced in panels (a) and (b) for the two higher resolutions, as the AIC waves are resolved and are dominant in the magnetosheath. A more detailed study about mirror modes in Vlasiator has been conducted by Hoilijoki et al. (2016).

The growth rate analysis shown in Fig. 6a suggests that the AIC waves are well resolved in the 300 km resolution run, at the highest resolution used in this study. Hoilijoki et al. (2016) showed that the mirror instability was also resolved in a different simulation.The observations of mirror modes in this study are consistent with the work of Hoilijoki et al. (2016). The results obtained with HYDROS show that the maximum growth rate of the proton cyclotron instability should be higher than that of the mirror instability in all three simulations, in agreement with previous numerical and theoretical studies (Price

et al., 1986; Gary, 1992; Gary et al., 1993; Gary and Winske, 1993; Shoji et al., 2009). The work of Shoji et al. (2009) and Shoji et al. (2012) show that the mirror instability becomes dominant when adding the third spatial dimension in hybrid-PiC simulations. Future 3D global hybrid-Vlasov simulations can be employed to further probe this balance question. The 600 km resolution case seems to limit the frequency of the waves below the ion cyclotron frequency, while still partially resolving the AIC waves, whereas the 900 km run highlights that only mirror modes are present in the magnetosheath and shows no

sign of the AIC waves, as evidenced by Fig. 2e-f and 3c, since the resolution of the simulation does not allow the Alfvén mode to grow sufficiently (Fig. 6). Remya et al. (2013) showed that an anisotropic electron distribution, which is not modeled in Vlasiator, with $T_{\perp,e}/T_{\parallel,e} > 1.2$ reduces the proton cyclotron instability growth rate, while increasing the mirror instability growth rate. In this case, the mirror instability will dominate over the proton cyclotron instability. Masood and Schwartz (2008) showed that the Earth's magnetosheath presents such anisotropic electron distributions. A later study by Ahmadi et al. (2016),

a comment by Remya et al. (2017) and a reply to this comment by Ahmadi et al. (2017) nuance this result. While Remya et al. (2013) conclusions hold true in the linear regime, Ahmadi et al. (2016) showed that, in the non-linear evolution, the anisotropic electron distributions will be unstable to the electron whistler instability. This instability, in the absence of heavy ions, will lower the anisotropy level and thus will restore the previous balance between the proton cyclotron instability and the mirror instability. To this day, there are no global-hybrid Vlasov simulations able to model both protons and anisotropic electrons in the Earth's magnetosheath to confirm these results. Fig. 5 of Soucek et al. (2015) also showed that the proton cyclotron waves should dominate in the magnetosheath for Mach numbers lower than 7. The Mach number in our simulation is $M_A = 6.9$. With all these factors considered, we expect both instabilities to be present in our simulations, and the proton cyclotron instability to have a higher maximum growth rate than the mirror instability in our simulations, when both modes are resolved. However, at the lowest resolution, the range of k-vectors the simulation is able to model is not large enough to allow the proton cyclotron instability to grow, regardless of which mode is expected to dominate.

McKean et al. (1994) showed that introducing Helium ions in a simulation tend to suppress the proton cyclotron instability (supported by Remya et al. (2013)), with only the mirror instability remaining. While there are no heavy ions in our study, our results display similar wave properties in the magnetosheath with a resolution of 900 km, when the proton cyclotron instability does not develop. The growth rate analysis of the mirror instability shown in Fig. 6b suggests that the mirror modes barely grow in the middle of the magnetosheath, where the data were taken, for $\Delta r = 300$ km. Hoilijoki et al. (2016) have shown in a different simulation that mirror-like structures are still present in the middle of the magnetosheath. Both kinetic instabilities grow near the quasi-perpendicular bow shock, where the temperature anisotropy is higher, and the resulting waves and structures propagate and travel with the plasma flow in the magnetosheath. The proton cyclotron instability however grows much faster and isotropises the ions (Davidson and Ogden, 1975; McKean et al., 1992). At $\Delta r = 600$ km, the mirror instability has a larger maximum growth rate than at $\Delta r = 300$ km, but still lower than that of the proton cyclotron instability. The proton cyclotron instability is more efficient to isotropise ions than the mirror instability (McKean et al., 1992), but cannot develop completely, hence the beginning of a loss-cone observed in Fig. 8f and Fig. 8h. At the lowest resolution, the spectrum of wave vectors triggered by the instabilities is not broad enough to scatter particles and thermalise the plasma. Therefore, we infer that no instability grows in the middle of the magnetosheath at this resolution.

The plasma $\beta$ is an important parameter to the development of these instabilities (Tsurutani et al., 1982; Gary, 1992). With different initial parameters, such as solar wind speed or IMF strength, the plasma $\beta$ in the magnetosheath would be different. One could then expect different results regarding the correlation between the spatial resolution and the instabilities. Using the plasma solver HYDROS, we investigate how the plasma $\beta$ affects the growth rate of the two instabilities (not shown). We found that an increased value of $\beta$ leads to a higher value of the maximum growth rate $\gamma_{\max}$, but does not change significantly the value of the corresponding wave vector $k$. A higher value of $\beta$ could slightly improve runs with resolution between 300 and 600 km. The low resolution run would still not resolve a large enough spectrum of wave vectors to allow the proton cyclotron instability to develop.

Panels (a)-(c) of Fig. 7 display the y-component (out of plane) of the magnetic field at the three different resolutions. The AIC waves are clearly present throughout the quasi-perpendicular part of the magnetosheath at the highest resolution, and

have completely disappeared at the lowest resolution, as previously predicted by Fig. 6. The intermediate resolution displays that the AIC waves are still present in the simulation, but at lower amplitude. Panels (d)-(f) of Fig. 7 displays the temperature anisotropy of the global simulation at the three different resolutions. The consequence of the absence of the AIC waves can be observed as a higher temperature anisotropy of the magnetosheath at the lowest resolution. Fig. 7 indicates that the temperature anisotropy grows larger as the spatial resolution of the simulation decreases. The AIC waves isotropize VDFs faster than the mirror modes, reducing the temperature anisotropy (Davidson and Ogden, 1975; McKean et al., 1992). Moreover, one can notice that the position of the quasi-perpendicular bow shock is more outward at the resolution $\Delta r = 900$ km than the others. Due to the AIC waves not being resolved, the temperature anisotropy is greatly increased behind the shock. The lack of mechanism to reduce the perpendicular pressure leads to an increase in the downstream perpendicular pressure, which further leads to a lower shock compression of the plasma. This in turn forces the shock to move outward faster. Therefore, a resolution higher than $\Delta r = 900$ km is necessary to model the quasi-perpendicular bow shock accurately.

The absence of the Alfvén mode at lower resolution leads to the discrepancies on the VDFs depending on the spatial resolution. Panels (b)-(d) of Fig. 8 show that, in the higher resolution case, the VDFs appear to have a nearly bi-Maxwellian shape, which is still partially present in panels (f)-(h) at $\Delta r = 600$ km, until this shape is deformed into an "hourglass" shape at the lowest resolution in panels (j)-(l). This shape suggests the presence of a loss-cone instability (Ichimaru, 1980), produced by mirror modes like structures. However, the growth rate of this instability is too slow to develop further. Particles cannot be scattered by the AIC waves, which are absent at low resolution, and hence the particles are trapped within the mirror modes. Therefore we observe a loss-cone in pitch-angle in the VDFs at this resolution. This loss-cone does not appear at the highest resolution, as the AIC waves dominate the wave-particle interaction when both instabilities are present (McKean et al., 1994). The VDFs shown in Fig. 8 are representative of those observed throughout the studied time range, as can be seen in supplementary videos.

Unresolved waves lead to energy transfer processes which are not being properly simulated, and hence lead to larger temperature anisotropies. One could argue that an easy way to get rid of this issue and to resolve physics beyond the ion inertial length (e.g. kinetic Alfvén waves) would be to use a higher spatial resolution. We conducted a similar study for a spatial resolution of $\Delta r = 227$ km (this simulation was also used in Hoilijoki et al. (2016)), which resolves the ion inertial length, and shows no evidence of new phenomena or wave modes, nor a better modelling of the ones already present at $\Delta r = 300$ km. High-resolution simulations are numerically costly, and therefore are not feasible globally in this fine resolution for the entire volume. This is especially true in large simulations such as global 6D simulations (3D real-space grid and 3D velocity-space grid) which will require adaptive mesh refinement allowing to focus resolution on regions of interest and decrease the resolution and computational cost significantly elsewhere. For our solar wind driving parameters, the ion inertial length is 227.7 km, and around 135.0 km in the magnetosheath for all simulations. We find that, despite not fully resolving the inertial length, the resolution $\Delta r = 300$ km leads to well resolved proton cyclotron and mirror instabilities. Since they are the two main competing instabilities in magnetosheath plasmas (Anderson and Fuselier, 1993; Gary, 1992; Soucek et al., 2015), we find that the resolution $\Delta r = 300$ km is sufficient to correctly resolve these waves in the magnetosheath. We also find that even at the intermediate resolution $\Delta r = 600$ km, the proton cyclotron instability still produces left-handed nearly circularly polarised

waves, and almost bi-Mawellian VDFs. While slightly anisotropic, the distributions reach marginal stability levels. Therefore
we believe that an acceptable minimum spatial resolution in a simulation to study magnetosheath waves would lie between
$\Delta r = 300$ and $600$ km.

It is evident that one can make a choice of spatial resolution depending on which waves are wanted in the simulation.
However, in case of large-scale simulation volumes where the entire simulation box cannot be represented with a uniform
grid resolution, it is interesting to contemplate whether one can use a sub-grid model to reproduce the most important wave
modes at the coarser grid volumes. A future topic of study would be to design an empirical model based on the results we have
presented here in this article, in order to modify the VDFs to a more Maxwellian shape, or to solve the Vlasov equation with
adding a diffusion term at lower resolution in order to mimic the energy dissipation mechanisms at work at smaller scales.

## 5   Conclusions

This paper presents an investigation into the spatial resolution dependence of two proton instabilities in a global hybrid-
Vlasov model. Three 2D simulations of the near Earth-space at different spatial grid resolutions are carried out, and the effects
on the produced magnetosheath waves and velocity distribution functions downstream of the quasi-perpendicular shock are
investigated.

The first simulation uses a resolution of $\Delta r = 300$ km $= 1.32$ $d_{i,SW} = 2.22$ $d_{i,M}$. The proton cyclotron instability is identified
by the production of left-hand nearly circularly polarised waves around the ion cyclotron frequency, with properties consistent
with those of Alfvén Ion Cyclotron waves. The VDFs have a nearly bi-Maxwellian shape, indicating isotropization of the
species. We also observe mirror modes in the middle of the magnetosheath, although with a lower growth rate, indicating that
they grow further upstream. This resolution allows the proton cyclotron and mirror instabilities to grow adequately.

The second simulation uses a resolution of $\Delta r = 600$ km $= 2.64$ $d_{i,SW} = 4.44$ $d_{i,M}$. The AIC waves are still present at this
resolution, yet not completely resolved. The VDFs are still nearly bi-Maxwellian, with a small loss-cone starting to appear,
due the fact that the AIC are only partially resolved. The temperature anisotropy is hence larger than at the previous resolution.
Even though the growth rate is larger than at $\Delta r = 300$ km, the resolution does not allow the maximum growth rate of the
proton cyclotron instability to be reached.

The third simulation uses a resolution of $\Delta r = 900$ km $= 3.96$ $d_{i,SW} = 6.67$ $d_{i,M}$. Large structures are observed, and the
VDFs display a significant loss-cone in the parallel directions. The anti-correlation of the fluctuations in magnetic field and
density highlights the presence of mirror modes. In this simulation the temperature anisotropy is much larger than at higher
resolutions. This is because the AIC waves are not present anymore. At this resolution, the spectrum of wave vectors is not
large enough to allow the instability to grow, regardless of which mode is expected to dominate. The predominance of wave
modes such as the proton cyclotron and mirror modes in the magnetosheath is an active topic of research.

This work shows that the proton cyclotron instability does not develop at low spatial resolution, assuming the plasma condi-
tions allow it to develop in the first place. In this case, energy dissipation processes are missing and thus the velocity distribution
functions are not isotropised. Larger simulations with inhomogeneous spatial resolution scale should include a sub-grid model,

like velocity space diffusion. This would account for the effects of the proton cyclotron instability without a significant increase of numerical resources.

The currently available runs allow us to conclude that the wave modes of interest here are properly resolved at a resolution of 300 km $= 1.32$ $d_{i,SW} = 2.22$ $d_{i,M}$. The growth rate profiles suggest that larger cell sizes, between 300 and 600 km, may still be sufficient to resolve those wave modes in the simulations. This hypothesis could be tested by running additional global simulations with a range of spatial resolutions. Such parametric study is however not currently achievable because of the large computational costs of global Vlasiator runs at these relatively high resolutions. At the highest resolution, the proton cyclotron

instability is well resolved. The proton cyclotron instability and the mirror instability are the two competing instabilities in this simulation. Based on our results, we conclude that, if the focus of the simulation is to evaluate the effects related to the proton cyclotron and mirror instabilities in the magnetosheath, there is no need to increase the spatial resolution of a simulation beyond $\Delta r = 300$ km $= 1.32$ $d_{i,SW} = 2.22$ $d_{i,M}$ at the cost of numerical resources.

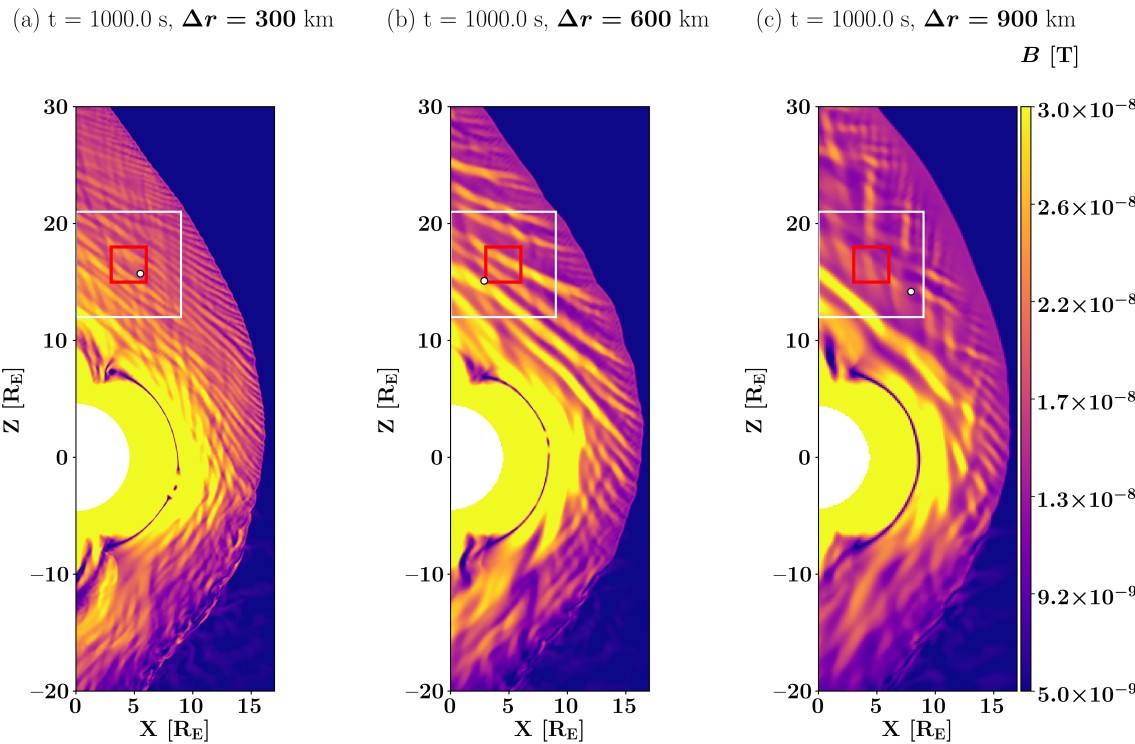

**Figure 1.** Global overview of the simulation setup with three different spatial resolutions: (a) $\Delta r = 300$ km, (b) $\Delta r = 600$ km and (c) $\Delta r = 900$ km. The colormap in each run is the magnitude of the magnetic field. The white square displays the area we focus on in the magnetosheath in the rest of the study. The red square displays the area where the FFT in Fig. 2 is performed. The black and white dot displays the location where data are taken for Fig. 3, 4, 5, 6, and the VDFs in Fig. 8.

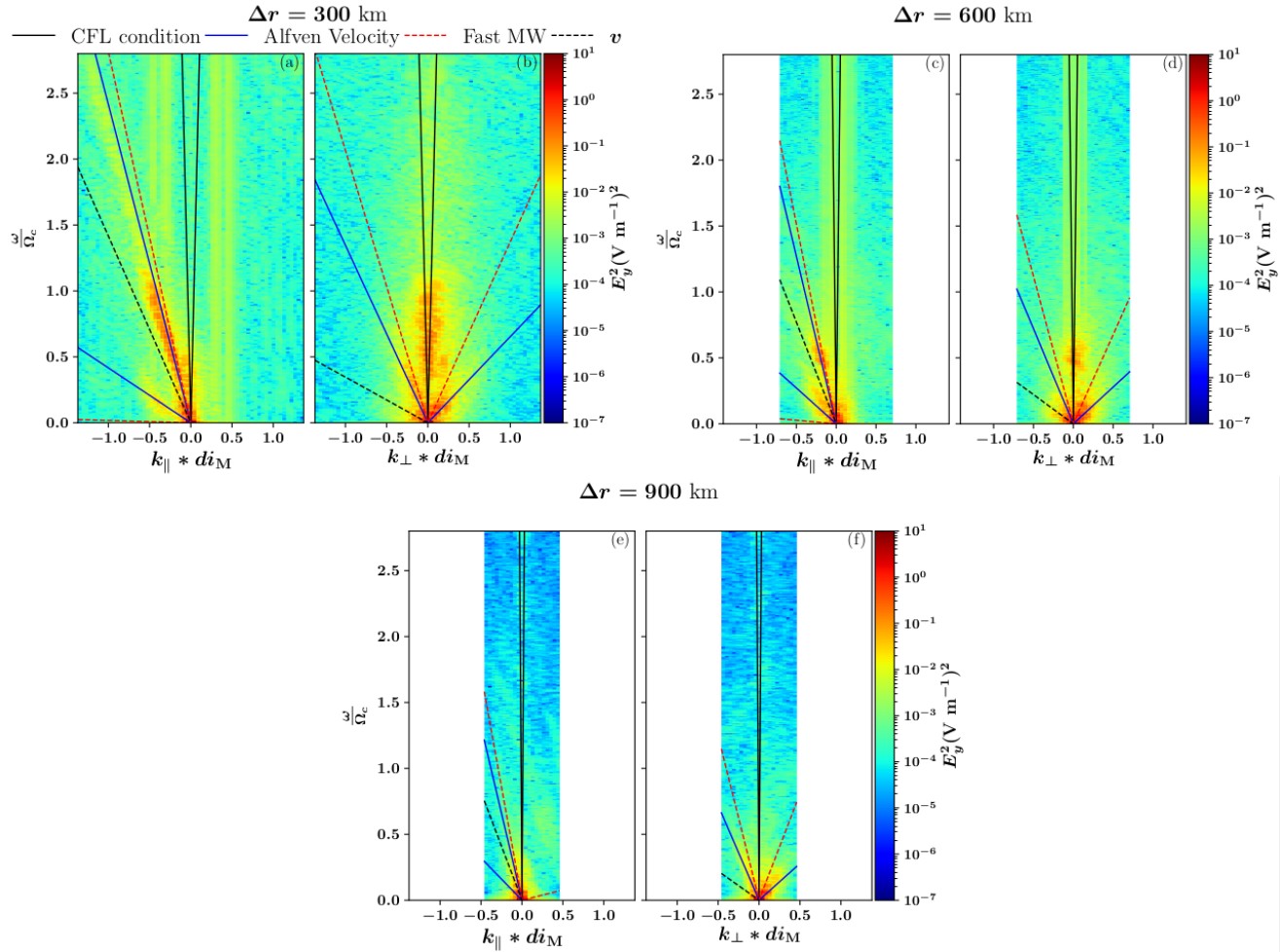

**Figure 2.** 2D Fast Fourier Transform of the y-component of the electric field in the direction parallel and perpendicular to the background magnetic field, at the location depicted by a red square in Fig. 1. Panels (a) and (b) display the results for the run at resolution $\Delta r = 300$ km, panels (c) and (d) the results for the run at resolution $\Delta r = 600$ km, and panels (e) and (f) the results for the run at resolution $\Delta r = 900$ km. The solid black lines represent the Courant-Friedrichs-Lewy condition, the solid blue lines the Alfvén speed, the dashed red lines the fast magnetosonic speed (labelled MW), with all wave frequencies shown in the simulation frame. The dashed black lines show the Doppler shift due to the plasma bulk flow.

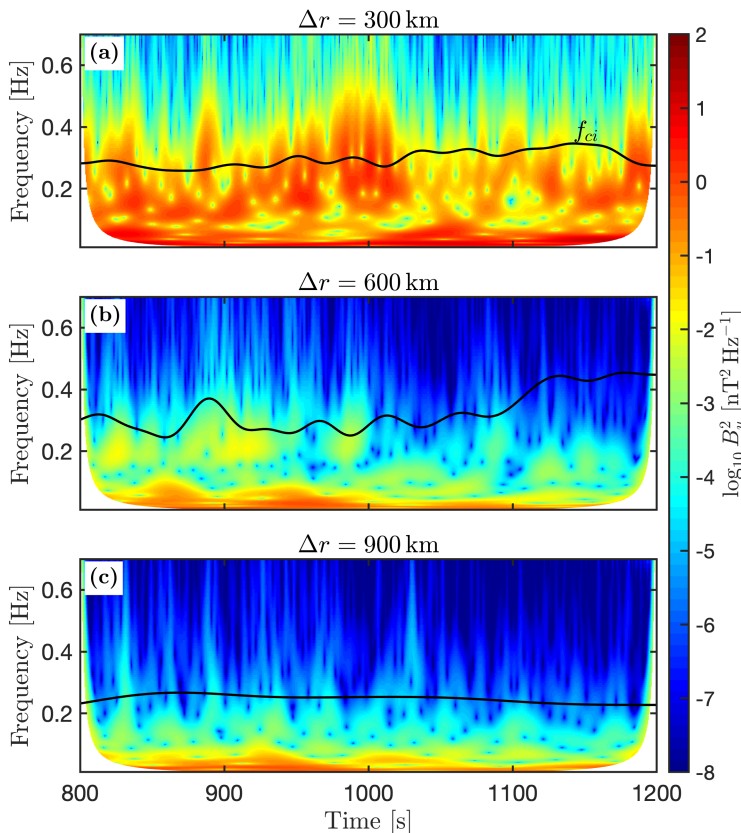

**Figure 3.** Wavelet analysis of the magnetic field for the virtual space craft locations given in Fig. 1. The colour background represents the power spectrum density of the y-component of the magnetic field. Panels (a) displays the results for the run at resolution $\Delta r = 300$ km, panel (b) the run at resolution $\Delta r = 600$ km, and panel (c) the run at resolution $\Delta r = 900$ km. The black curve on each plot indicates the proton cyclotron frequency.

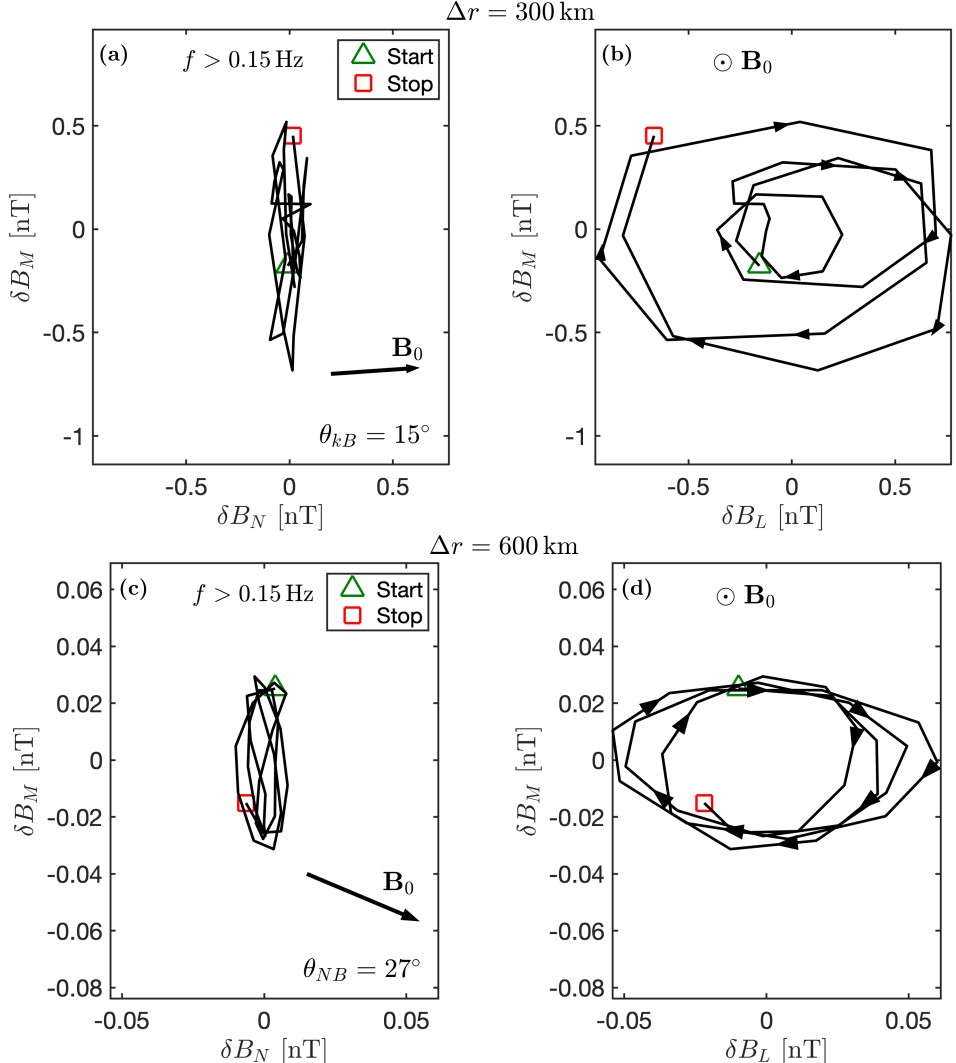

**Figure 4.** Hodogram of the magnetic field fluctuations. Panels (a) and (b) display the case $\Delta r = 300$ km, taken between $t = 996.5$ s and $t = 1013.5$ s. Panels (c) and (d) display the case $\Delta r = 600$ km, taken between $t = 910.0$ s and $t = 928.0$ s. Panels (a) and (c): intermediate ($\delta B_M$) and minimum ($\delta B_N$) variance directions. An arrow marks the average background field $\mathbf{B}_0$. Panels (b) and (d): intermediate ($\delta B_M$) and maximum ($\delta B_L$) variance directions. Arrows show the time evolution of the fluctuations. A green triangle marks the start of the interval and a red square marks the end.

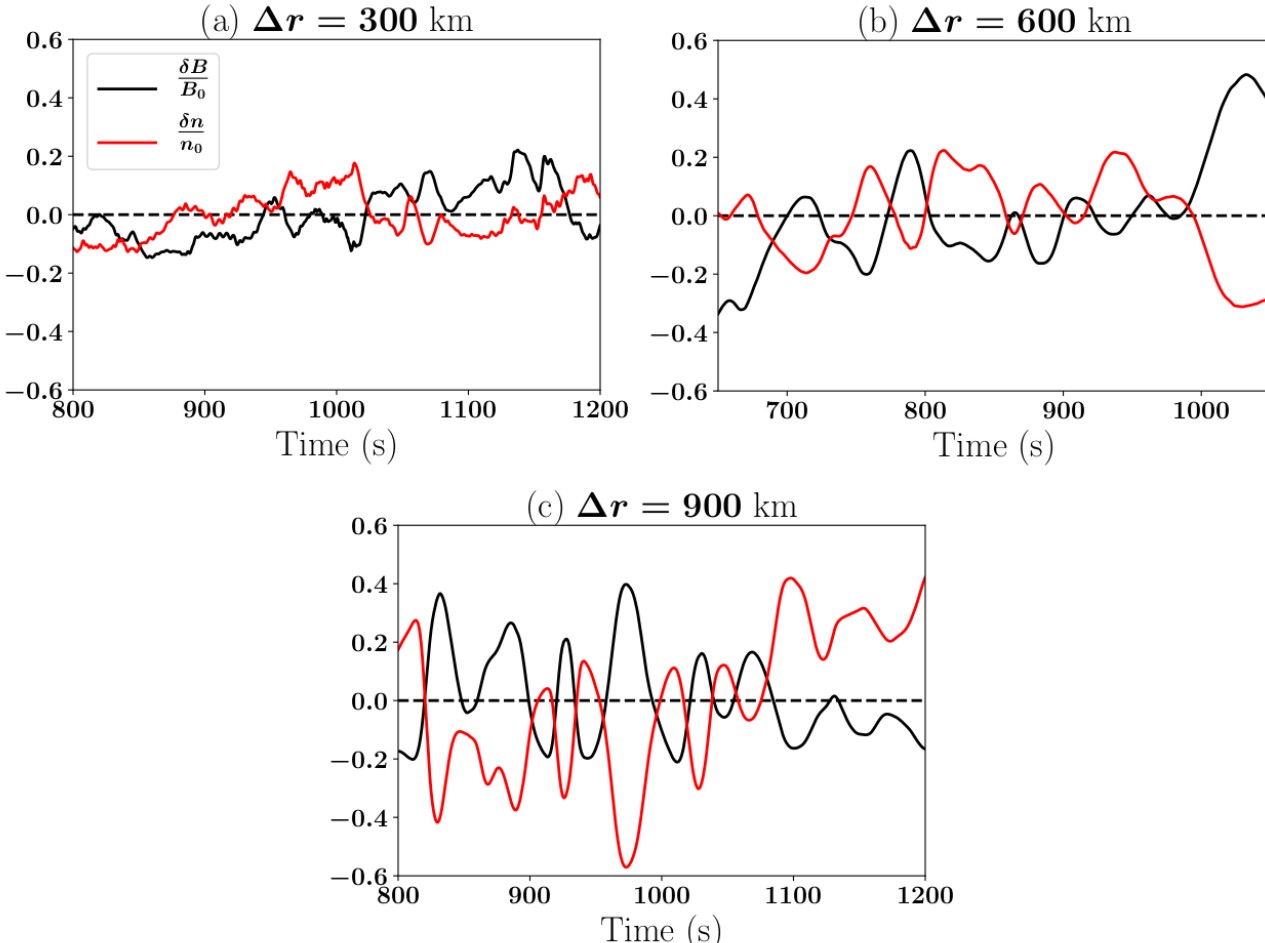

**Figure 5.** Magnetic field (black) and density (red) fluctuations for the three simulations measured at the virtual spacecraft locations indicated by a black and white dot in Fig. 1.

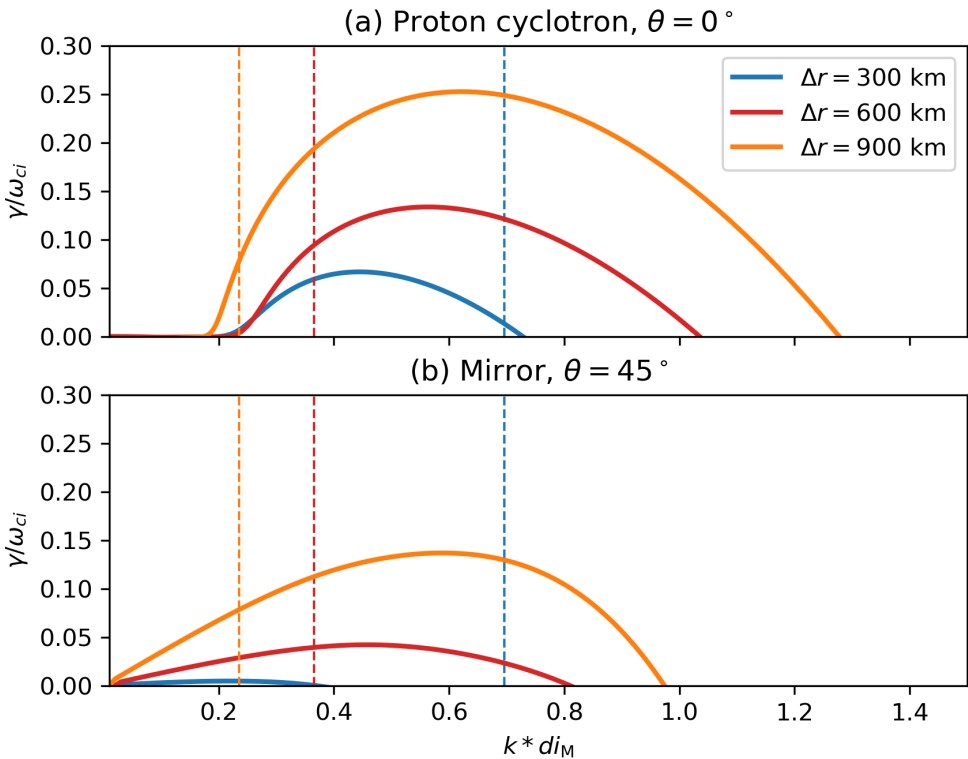

**Figure 6.** Growth rates $\gamma$ of the proton cyclotron (panel (a)) and Mirror (panel (b)) instabilities calculated by HYDROS for the three different spatial resolutions: $\Delta r = 300$ km (blue), $\Delta r = 600$ km (red) and $\Delta r = 900$ km (orange). The dashed lines represent the maximum wave number $k_{\mathrm{max}} = \pi/2\Delta r$ which can be resolved by the model to get a proper description of the signal at each resolution.

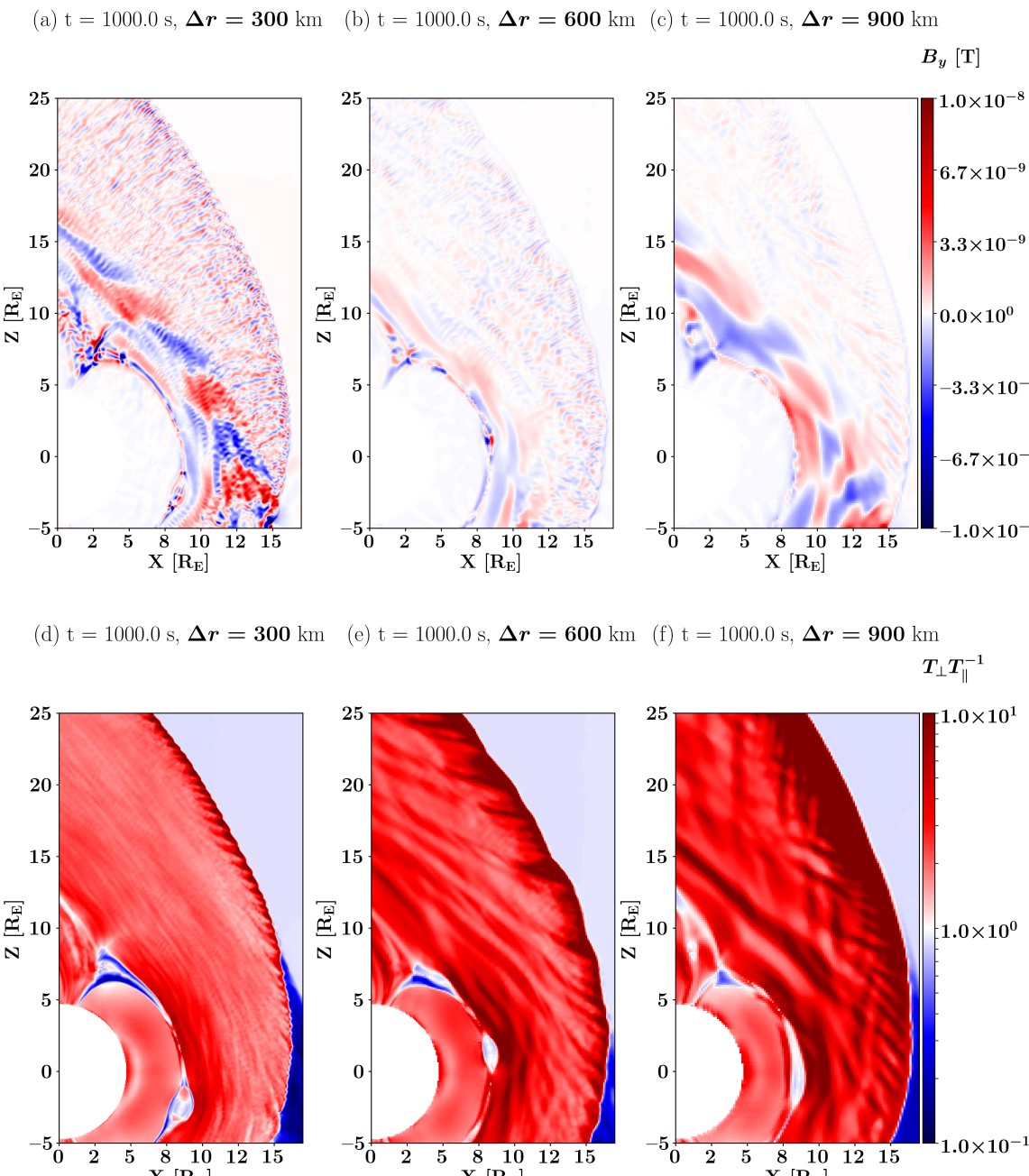

**Figure 7.** Y-component of the magnetic field for the resolution: (a) $\Delta r = 300$ km, (b) $\Delta r = 600$ km, (c) $\Delta r = 900$ km, and temperature anisotropy for the resolution: (d) $\Delta r = 300$ km, (e) $\Delta r = 600$ km, (f) $\Delta r = 900$ km.

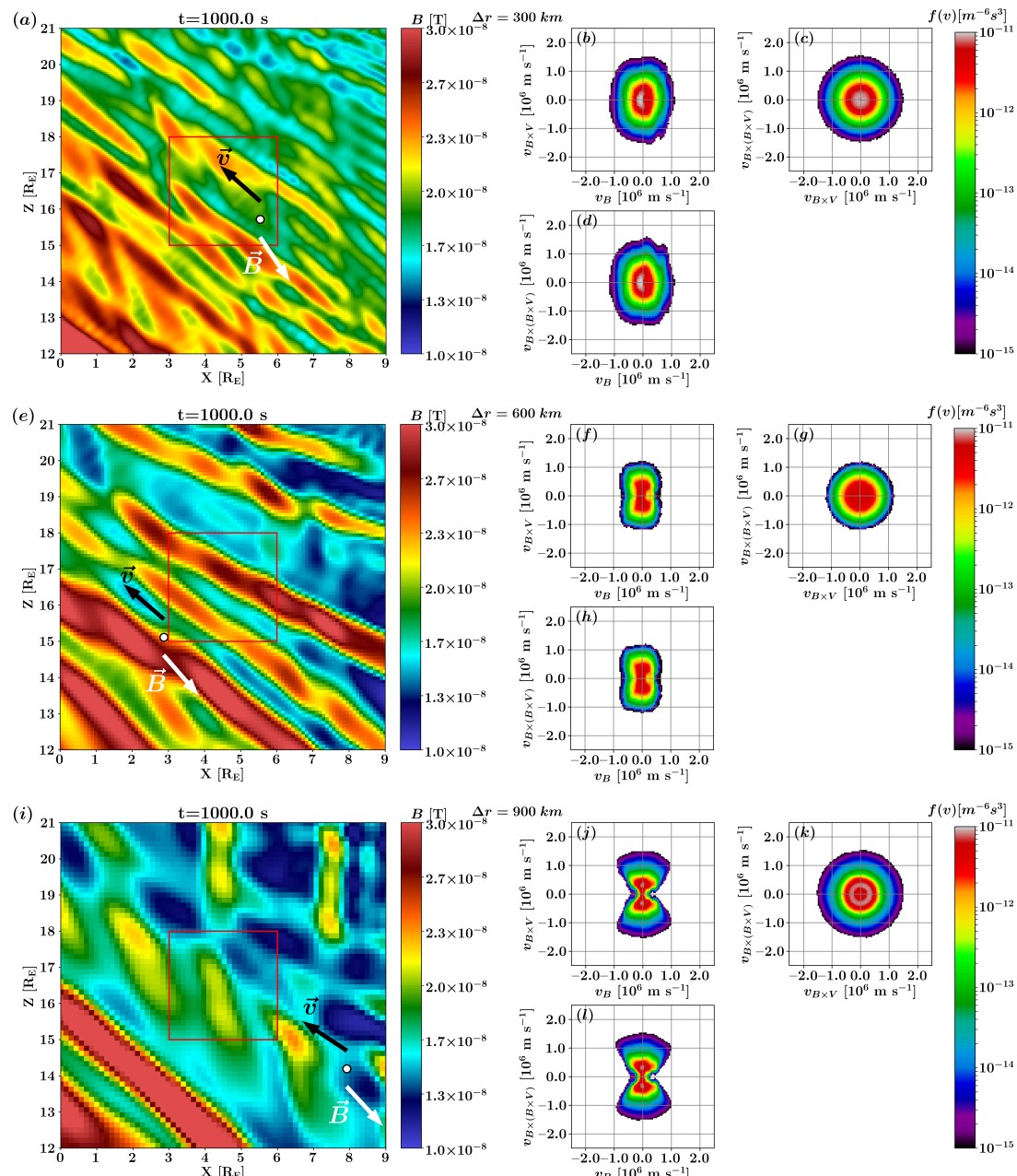

**Figure 8.** Colormap of the magnetic field (left) and velocity distribution functions in the three directions (right) located at the black and white circle. Panels (a)-(d) display the run with resolution $\Delta r = 300$ km, panels (e)-(h) the run with resolution $\Delta r = 600$ km and panels (i)-(l) the run with resolution $\Delta r = 900$ km. The black arrow displays the plasma bulk velocity and the white arrow displays the magnetic field direction, both taken at the location indicated by the black and white circle. The red square displays the area where the 2D-FFT is taken (Fig. 2).

**Table 1.** Summary table for each run with spatial resolutions, the inertial length $d_{i,SW}$ in the solar wind, the inertial length $d_{i,M}$ in the magnetosheath, the maximum wave vector allowed by the simulation $k_{max} = \pi/\Delta r$, the wave vector at which the proton cyclotron instability's growth rate is maximum $k_{\gamma_{max}}$, status of the proton cyclotron and mirror instabilities in the simulation, VDF shapes in the ($\mathbf{v_B}$, $\mathbf{v_{B} \times v}$) plane (Fig. 8b, f and j), temperature anisotropy, and plasma beta taken at the location indicated in Fig. 1 averaged over the time range used in this study.

| $\Delta r$ (km) | $d_{i,SW}$ (km) | $d_{i,M}$ (km) | $k_{max}d_{i,M}$ | $k_{\gamma_{max}}d_{i,M}$ | Proton cyclotron | Mirror | VDF ($\mathbf{v_B}$, $\mathbf{v_{B} \times v}$) | $T_\perp/T_\parallel$ | $\beta$ |
|---|---|---|---|---|---|---|---|---|---|
| 300 | 228 | 135 | 0.68 | 0.44 | yes | yes | Bi-Maxwellian | 1.97 | 2.58 |
| 600 | 228 | 135 | 0.35 | 0.53 | partially | yes | Nearly bi-Maxwellian beginning of loss-cone | 2.56 | 2.72 |
| 900 | 228 | 135 | 0.23 | 0.63 | no | yes | Loss-cone | 3.41 | 4.18 |

*Code and data availability.* Vlasiator (http://www.physics.helsinki.fi/vlasiator/, (Palmroth et al., 2018) is distributed under the GPL-2 open source license at https://github.com/fmihpc/vlasiator/ (Palmroth & the Vlasiator team, 2019). Vlasiator uses a data structure developed in-house (https://github.com/fmihpc/vlsv/, Sandroos, 2018), which is compatible with the VisIt visualization software (Childs et al., 2012) using a plugin available at the VLSV repository. The Analysator software (https://github.com/fmihpc/analysator/, Hannuksela & the Vlasiator team, 2018) was used to produce the presented figures. The runs described here take several terabytes of disk space and are kept in storage maintained within the CSC – IT Center for Science. Data presented in this paper can be accessed by following the data policy on the Vlasiator website.

*Video supplement.* Video supplements are available at https://doi.org/10.5446/46345, https://doi.org/10.5446/46730 and https://doi.org/10.5446/46731

*Author contributions.* MD carried out most of the study and the writing of the paper. UG and YP-K participated in running the different simulations and the development of the analysis methods. MB participated in the development of the analysis methods. AO, MG, AJ and LT contributed in the data analysis and discussions. MP is the PI of the Vlasiator model and gave input on the interpretation of the simulation results. All co-authors participated in the discussion of the results and contributed in improving the manuscript.

*Competing interests.* The authors declare they have no conflicts of interest.

*Acknowledgements.* We acknowledge The European Research Council for Starting grant 200141-QuESpace, with which Vlasiator (Palmroth et al., 2018) was developed, and Consolidator grant 682068-PRESTISSIMO awarded to further develop Vlasiator and use it for scientific investigations. The Finnish Centre of Excellence in Research of Sustainable Space (FORESAIL), funded through the Academy of Finland grant number 312351, supports Vlasiator development and science as well. The CSC – IT Center for Science in Finland is acknowledged for the Sisu and Taito supercomputer usage, which led to the results presented here.

Maxime Dubart thanks Thiago Brito for useful discussions throughout the study which helped carrying it out.

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
