# Peer review of "Resolution dependence of magnetosheath waves in global hybrid-Vlasov simulations"

_Annales Geophysicae, 2020_

## Referee Comment (RC1) · Anonymous Referee #1 · 27 May 2020

Review of "Resolution dependence of magnetosheath waves in global hybrid-Vlasov simulations" by Dubart, Ganse, Osmane, Johlander, Battarbee, Grandin, Pfau-Kempf, Turc and Palmroth The authors have written an article about a long-standing problem of whether mirror modes or ion cyclotron waves should dominate in the Earth's magnetosheath. This topic has a long history and there have been many papers (which were not referenced) on this topic. However more importantly, there have been some recent works that appear to have solved the problem. These papers unfortunately supercede the work that you have done. I therefore cannot accept your work to AG in anything resembling the present version. I will try to guide you to do some reading to catch up on the topic and perhaps redo your work to make a contribution to the literature. Major Comments: Two papers which address this issue of mirror

mode/ion cyclotron waves are: JGR, 114, A10203, doi:10.1029/2008JA014038, 2009 and JGRSP, 118, 785-793, 2013. Then a paper critical to the 2013 paper: JGR, 121, 5350–5365,doi:10.1002/2016JA022429, 2016. This was followed by a comment and reply: JGRSP, 122, 745–747, doi:10.1002/2016JA023148, 2017; JGRSP, 122, 748–752, doi:10.1002/2016JA023452, 2017.

The authors should also cite some of the earlier papers quoted in the 2009 and 2013 papers that address the issue of MM/EMIC growth rates. This will give more context to your work. These earlier papers state that the growth rate of the ion cyclotron wave is higher than the MM waves, contrary to your results. You will need to reconcile this point as well.

Minor Comments. The second sentence in the abstract may be incorrect. Please note that there are electron cyclotron waves in the magnetosheath called lion roars. Three references to this mode are: JGR, 81, 2261, 1976; JGR, 103, A3, 4615-4626, 1998; AnnGeo, 17, 1528-1534, 1999. Abstract lines 5-6. The authors should realize that ion cyclotron waves have been detected in the Earth's magnetosheath, perhaps under the right plasma circumstances: in Space Weather Study Using Multipoint Techniques, edited by L.-H. Lyu, Pergamon, 97, 2002. So the plasma conditions are necessary to take under consideration to determine what wave mode will grow. Two recent reviews on MM waves are: NPG, 17, 467-479, 2010 and JGR, 116, A02103, doi:10.1029/2010JA015913, 2011. Line 29. MM waves have also been detected at Jupiter's and Saturn's magnetosheaths as well. There are many references to this, including the original 1982 paper. Line 30: See introduction of 2011 JGR paper for more references to MM detection in interplanetary space (solar wind), in the Earth's geotail, and at comets. MMs have also been detected at heliopause. These references should be included. Line 255. I suggest that you add more references to some of the earlier work that focused on the issue of the growth of the two competing modes. This was mentioned earlier.

---

## Referee Comment (RC2) · Anonymous Referee #2 · 29 May 2020

The authors applied three 2-D global hybrid-Vlasov simulations to investigate the dependence of ion cyclotron and mirror mode instability on the spatial resolution. By comparing three runs, the authors conclude that $\Delta r \sim 0.6$ di is an acceptable minimum spatial resolution for a simulation to study magnetosheath waves. The importance of this work is to help future simulations to save resources. However, I have some concerns about the conclusion.

Major comments: Plasma $\beta$ is one very important parameter to two instabilities. If different solar wind speed, solar wind temperature, and IMF field strength are used in the simulation, the plasma $\beta$ in the magnetosheath will be very different. So my concern is that the concluded spatial resolution very likely depends on the magnetosheath plasma $\beta$ or solar wind parameters. Therefore, it is necessary to justify that the conclusion is

true for all the possible solar wind parameters or how the conclusion depends on the solar wind parameters. Otherwise, the importance of this work to future simulations will be very limited.

Under this certain solar wind condition, the authors conclude that $\Delta r \sim 440$ km = 0.6 di would be adequate. However, I am not convinced by this number which is based on the growth rate profile. For example, where the growth rate is calculated may not be the source region. So I wonder whether the authors can run several more cases, e.g., with $\Delta r$ around 400 km, to show that the results are indeed close enough to the case with $\Delta r$=300 km.

Minor comments: Please rephrase "magnetosheath waves" in the abstract and conclusion as there are not just mirror mode and AIC waves in the magnetosheath.

Why the position of quasi-perpendicular bow shock in Figure 1c is more outward than Figures 1a and 1b?

In Figure 2f, there are signals along two blue lines. What are they?

In Figure 5, it would be better if there are similar panels of other two runs for comparison.

There are some typos such as line 221, "were the data were taken" -> "where the data were taken" and line 225, "more efficient tp" -> "more efficient to".

---

## Author Comment (AC1) · 2 Jun 2020

**We thank the reviewer for their examination of our manuscript and their comments. Please find below our response to your concerns.**

"The authors have written an article about a long-standing problem of whether mirror modes or ion cyclotron waves should dominate in the Earth's magnetosheath. This topic has a long history and there have been many papers(which were not referenced) on this topic. However more importantly, there have been some recent works that appear to have solved the problem. These papers unfortunately supercede the work that you have done. I therefore cannot accept your work to AG in anything resembling

the present version. [...] The authors should also cite some of the earlier papers quoted in the 2009 and 2013 papers that address the issue of MM/EMIC growth rates. This will give more context to your work."

**We thank the reviewer for pointing out the lacking connections to previous work about the question of mirror mode versus ion cyclotron waves in the magnetosheath and for providing valuable references about this topic. We would like to emphasise that the intention of this paper is primarily to help future global hybrid-Vlasov simulations to save resources when simulating and investigating magnetosheath waves, and secondarily to investigate the mirror mode waves and the ion cyclotron waves in term of those resolutions, not the competition between these modes. We will add the suggested references in the revised manuscript and revise the manuscript in terms of the physics the reviewer suggested. We will also add more clarity and emphasis to the primary focus of the paper.**

"These earlier papers state that the growth rate of the ion cyclotron wave is higher than the MM waves, contrary to your results. You will need to reconcile this point as well."

**Fig. 6 of our manuscript shows the growth rate of the proton cyclotron (Panel a) and mirror (Panel b) instabilities at the three different resolutions: 300 km (blue), 600 km (red), 900 km (orange). As can be seen, the proton cyclotron growth rate is higher than the mirror growth rate in all resolutions, consistent with the previous works mentioned by the reviewer.**

"The second sentence in the abstract may be incorrect. Please note that there are electron cyclotron waves in the magnetosheath called lion roars."

**We will emphasise in the revised manuscript that we focus on ion-scale waves.**

"MM waves have also been detected at Jupiter's and Saturn's magnetosheaths as well. There are many references to this, including the original 1982 paper. Line 30: See introduction of 2011 JGR paper for more references to MM detection in interplanetary space (solar wind), in the Earth's geotail,and at comets. MMs have also been detected at heliopause. These references should be included. Line 255. I suggest that you add more references to some of the earlier work that focused on the issue of the growth of the two competing modes. This was mentioned earlier."

**Thank you very much for pointing this out. We will gladly provide more references about the detection of these waves in these other setups.**

---

## Author Comment (AC2) · 5 Jun 2020

**We thank the reviewer for their careful examination of our manuscript and their constructive comments. Please find below our response to your concerns.**

"Major comments: Plasma $\beta$ is one very important parameter to two instabilities. If different solar wind speed, solar wind temperature, and IMF field strength are used in the simulation, the plasma $\beta$ in the magnetosheath will be very different. So my concern is that the concluded spatial resolution very likely depends on the magnetosheath plasma $\beta$ or solar wind parameters. Therefore, it is necessary to justify that the conclusion is true for all the possible solar wind parameters or how the

[Figure]

conclusion depends on the solar wind parameters. Otherwise, the importance of this work to future simulations will be very limited."

**This is an excellent point we will add to the discussion. While running an entire new set of simulations with different solar wind parameters would be too expensive, we plotted the growth rate of the proton cyclotron instability with three different $\beta$ (the reviewer can find the figure attached to this document). An increased plasma $\beta$ gives a higher value of the maximum growth rate $\gamma_{\text{max}}$, but does not change significantly the value of the corresponding wave vector $k$. Our conclusions remain that the low resolution run does not resolve a large enough spectrum of wave vectors to allow the proton cyclotron instability to develop. A higher beta value would slightly improved runs with resolution between 300 and 600 km. We will add this to the discussion in the revised manuscript.**

"Under this certain solar wind condition, the authors conclude that $\Delta r \approx 440$ km = 0.6 di would be adequate. However, I am not convinced by this number which is based on the growth rate profile. For example, where the growth rate is calculated may not be the source region. So I wonder whether the authors can run several more cases, e.g.,with $\Delta r$ around 400 km, to show that the results are indeed close enough to the case with $\Delta r = 300$ km"

**We thank the reviewer for pointing this out. We agree that this conclusion, purely made based on the growth rate profile, is probably too specific, since many other parameters, such as the source region of the wave in the global simulation, may affect it. As mentioned earlier, running additional Vlasiator simulations is difficult because of their large computational cost (a few million CPU hours for a 400 km run similar as those presented in this study). In the revised manuscript, we will reformulate our conclusions as follows:**

The currently available runs allow us to conclude that the wave modes of interest here are properly resolved at a resolution of 300 km = 1.32 di. The growth rate profiles suggest that larger cell sizes, between 300 and 600 km, may still be sufficient to resolve those wave modes in the simulations. This hypothesis could be tested in running additional global simulations with a range of spatial resolutions. Such a parametric study is however not currently achievable because of the large computational costs of global Vlasiator runs at these relatively high resolutions.

Incidentally, we have realised that there was an error with the value of the resolution in term of ion skin depth in the manuscript. The values should be $\Delta r = 300$ **km** $= 1.32\ d_i$, $\Delta r = 600$ **km** $= 2.64\ d_i$, $\Delta r = 900$ **km** $= 3.96\ d_i$ **and** $\Delta r = 400$ **km** $= 1.76\ d_i$, as the ion skin depth in the solar wind is 228 km. This does not however affect our conclusions. We will correct this in the revised manuscript.

"Minor comments: Please rephrase "magnetosheath waves" in the abstract and conclusion as there are not just mirror mode and AIC waves in the magnetosheath."

We will rephrase this formulation in the revised manuscript.

"Why the position of quasi-perpendicular bow shock in Figure 1c is more outward than Figures 1a and 1b?"

We thank the reviewer for raising this discussion. As can be seen on Fig. 7, due to the AIC waves not being resolved, the temperature anisotropy is greatly increased in the 900 km run. The lack of mechanism to reduce the perpendicular

temperature leads to an increased pressure behind the shock, "pushing" it further away. We will add this to the discussion in the revised manuscript as an additional argument toward the importance of setting the resolution to 300 km.

"In Figure 2f, there are signals along two blue lines. What are they?"

**We believe it to be due to some numerical artefacts. The low resolution and the fact that these features appear in the perpendicular direction make us doubt that this would be related to Alfven waves. We currently apply a Hanning window filtering in both time and spatial dimension. We are planning to add other filters and see if these features disappear.**

"In Figure 5, it would be better if there are similar panels of other two runs for comparison."

**We will add them in the revised manuscript.**

"There are some typos such as line 221, 'were the data were taken' → 'where the datawere taken' and line 225, 'more efficient tp' → 'more efficient to'".

**We thank the reviewer for picking up these typos, we will correct them in the revised manuscript.**

[Figure]

**Fig. 1.** Growth rate of the proton cyclotron instability for three different beta plasma, computed with HYDROS

[Figure]

---

## Author Response (AR1)

**Response to reviewers**

**Response to Reviewer #1**

**We thank the reviewer for their examination of our manuscript and their comments. Please find below, in bold, our response to your concerns, and the changes added to the revised manuscript. The lines mentioned are pointing to the manuscript with tracked changes, that the reviewer can find at the end of the responses.**

"The authors have written an article about a long-standing problem of whether mirror modes or ion cyclotron waves should dominate in the Earth's magnetosheath. This topic has a long history and there have been many papers(which were not referenced) on this topic. However more importantly, there have been some recent works that appear to have solved the problem. These papers unfortunately supercede the work that you have done. I therefore cannot accept your work to AG in anything resembling the present version. [...] The authors should also cite some of the earlier papers quoted in the 2009 and 2013 papers that address the issue of MM/EMIC growth rates. This will give more context to your work."

**We thank the reviewer for pointing out the lacking connections to previous work about the question of mirror mode versus ion cyclotron waves in the magnetosheath and for providing valuable references about this topic. We would like to emphasise that the intention of this paper is primarily to help future global hybrid-Vlasov simulations to save resources when simulating and investigating magnetosheath waves, and secondarily to investigate the mirror mode waves and the ion cyclotron waves in term of those resolutions, not the competition between these modes. We have added the suggested references in the revised manuscript and revised the manuscript in terms of the physics the reviewer suggested (Lines 50 to 54). We have added more clarity and emphasis to the primary focus of the paper (Abstract).**

"The second sentence in the abstract may be incorrect. Please note that there are electron cyclotron waves in the magnetosheath called lion roars."

**We have emphasised in the revised manuscript that we focus on**

**ion-scale waves (Abstract and Lines 50 to 54).**

"These earlier papers state that the growth rate of the ion cyclotron wave is higher than the MM waves, contrary to your results. You will need to reconcile this point as well."

**Fig. 6 of our manuscript shows the growth rate of the proton cyclotron (Panel a) and mirror (Panel b) instabilities at the three different resolutions: 300 km (blue), 600 km (red), 900 km (orange). As can be seen, the proton cyclotron growth rate is higher than the mirror growth rate in all resolutions, consistent with the previous works mentioned by the reviewer. We have added references the reviewer provided to compare our results to previous work (Lines 249 to 258).**

"MM waves have also been detected at Jupiter's and Saturn's magnetosheaths as well. There are many references to this, including the original 1982 paper. Line 30: See introduction of 2011 JGR paper for more references to MM detection in interplanetary space (solar wind), in the Earth's geotail,and at comets. MMs have also been detected at heliopause. These references should be included. Line 255. I suggest that you add more references to some of the earlier work that focused on the issue of the growth of the two competing modes. This was mentioned earlier."

**Thank you very much for pointing this out. We have provided more references about the detection of these waves in these other setups and to emphasise previous work on the topic (Lines 39 to 52 and 57 to 67).**

**Response to Reviewer #2**

**We thank the reviewer for their careful examination of our manuscript and their constructive comments. Please find below, in bold, our response to your concerns, and the changes added to the revised manuscript. The lines mentioned are pointing to the manuscript with tracked changes, that the reviewer can find at the end of the responses.**

"Major comments: Plasma $\beta$ is one very important parameter to two instabilities. If different solar wind speed, solar wind temperature, and IMF field strength are used in the simulation, the plasma $\beta$ in the magnetosheath will be very different. So my concern is that the concluded spatial resolution very likely depends on the magnetosheath plasma $\beta$ or solar wind parameters. Therefore, it is necessary to justify that the conclusion is true for all the possible solar wind

parameters or how the conclusion depends on the solar wind parameters. Otherwise, the importance of this work to future simulations will be very limited."

**This is an excellent point. While running an entire new set of simulations with different solar wind parameters would be too expensive, we plotted the growth rate of the proton cyclotron instability with three different $\beta$ (the reviewer can find the figure attached to this document). An increased plasma $\beta$ gives a higher value of the maximum growth rate $\gamma_{\mathrm{max}}$, but does not change significantly the value of the corresponding wave vector $k$. Our conclusions remain that the low resolution run does not resolve a large enough spectrum of wave vectors to allow the proton cyclotron instability to develop. A higher beta value would slightly improved runs with resolution between 300 and 600 km. We have added this to the discussion in the revised manuscript (Lines 276 to 283).**

"Under this certain solar wind condition, the authors conclude that $\Delta r \approx$ 440 km = 0.6 di would be adequate. However, I am not convinced by this number which is based on the growth rate profile. For example, where the growth rate is calculated may not be the source region. So I wonder whether the authors can run several more cases, e.g.,with $\Delta r$ around 400 km, to show that the results are indeed close enough to the case with $\Delta r = 300$ km"

**We thank the reviewer for pointing this out. We agree that this conclusion is purely made based on the growth rate profile is probably too specific, since many other parameters, such as the source region of the wave in the global simulation, may affect it. As mentioned earlier, running additional Vlasiator simulations is difficult because of their large computational cost (a few million CPU hours for a 400 km run similar as those presented in this study). In the revised manuscript, we have reformulated our conclusions as follows (Abstract and Lines 354 to 358):**

**The currently available runs allow us to conclude that the wave modes of interest here are properly resolved at a resolution of 300 km = 1.32 di. The growth rate profiles suggest that larger cell sizes, between 300 and 600 km, may still be sufficient to resolve those wave modes in the simulations. This hypothesis could be tested in running additional global simulations with a range of spatial resolutions. Such a parametric study is however not currently achievable because of the large computational costs of global Vlasiator runs at these relatively high resolutions.**

**Incidentally, we have realised that there was an error with the value of the resolution in term of ion skin depth in the manuscript. The values should be $\Delta r = 300$ km $= 1.32\ d_i$, $\Delta r = 600$ km $= 2.64\ d_i$, $\Delta r = 900$ km $= 3.96\ d_i$ and $\Delta r = 400$ km $= 1.76\ d_i$, as the ion skin**

**depth in the solar wind is 228 km. This does not however affect our conclusions. This has been corrected throughout the masnucript.**

"Minor comments: Please rephrase "magnetosheath waves" in the abstract and conclusion as there are not just mirror mode and AIC waves in the magnetosheath."

**We have rephrased to focus on ion-scale instabilities in the abstract (Line 3) and the in conclusions (Lines 360 to 362). We have also clarified our conclusions regarding the availability of other runs (Lines 354 to 358).**

"Why the position of quasi-perpendicular bow shock in Figure 1c is more outward than Figures 1a and 1b?"

**We thank the reviewer for raising this discussion. As can be seen on Fig. 7, due to the AIC waves not being resolved, the temperature anisotropy is greatly increased in the 900 km run. The lack of mechanism to reduce the perpendicular temperature leads to an increased pressure behind the shock, "pushing" it further away. We have added this to the discussion (Lines 298 to 303) in the revised manuscript as an additional argument toward the importance of setting the resolution to 300 km.**

"In Figure 2f, there are signals along two blue lines. What are they?"

**We believe it to be due to some numerical artefacts. The low resolution and the fact that these features appear in the perpendicular direction make us doubt that this would be related to Alfven waves. We currently apply a Hanning window filtering in both time and spatial dimension. We have tested other filters and the features are not disappearing.**
"In Figure 5, it would be better if there are similar panels of other two runs for comparison."

**We have updated Fig. 5 as requested, and added the related discussion (Lines 182 to 184).**

"There are some typos such as line 221, 'were the data were taken' → 'where the datawere taken' and line 225, 'more efficient tp' → 'more efficient to'".

**We thank the reviewer for picking up these typos, we have corrected them.**

**Relevant changes made to manuscript**

**Abstract**

- Added and modified a few sentences to clarify the goal of the paper and to highlight the focus on ion-scale waves.

**Introduction**

- Added references provided by the reviewers.

- Added a paragraph citing additional previous work on the topic of the paper.

- Added a few sentences to clarify the goal of the paper.

**Results**

- Added two panels to Figure 5. Added the related description in section 3.1.

**Discussion**

- Added the discussion related to Figure 5.

- Added a paragraph comparing the HYDROS results to previous studies and their agreement with our work.

- Added a paragraph discussing the relevance of the $\beta$ parameter in the study.

- Added a paragraph discussing the influence of the temperature anisotropy on the position of the bowshock.

**Conclusions**

- Modified the conclusion regarding the acceptable limits of the resolution.

**Other relevant changes**

- Corrected the values of the resolution in term of ion inertial length throughout the manuscript.

- Added two notations to clarify the difference between the ion inertial length in the solar wind and in the magnetosheath. Added a column in Table 1 for the additional notation.

**Tracked changes Manuscript**

Following is the revised manuscript. The discussions and minor revisions added for clarity can be found in bold font. The sentences removed from the manuscript can be found crossed.

[revised manuscript text omitted]

---

## Referee Report (RR1)

The authors have addressed all my concerns, so I would like to recommend the paper for publication. I only have one suggestion. Is it possible to generalize your concluded spatial resolution from Earth's magnetosheath to other plasma environments, e.g., foreshock and solar wind? Some discussion may help increase the impact of the study.

---

## Author Response (AR3)

**Response to reviewer #1**

We thank the reviewer for their second examination of our manuscript and their constructive comments. Please find below our response to your concerns, in bold font. The lines indicated in our response are pointing to the lines in the revised manuscript with tracked changes. The revised manuscript with tracked changes has also been added at the end of this document.

First of all from the previous set of comments that I made, if the Remya et al. 2013 prediction that electron temperature anisotropies are important for the reduction of the ion cyclotron growth rate, then your results will not match reality. This point should be made clear to the readership.

**We thank the reviewer for raising this discussion. Remya et al. (2013) showed that the inclusion of anisotropic electrons with $T_\perp/T_\parallel > 1.2$ reduces the ion cyclotron growth rate and increases the mirror mode growth rate, in the linear regime. Masood and Schwartz (2008) showed that the electrons exhibit such anisotropies in the magnetosheath. A critical paper by Ahmadi et al. (2016), followed by a comment by Remya et al. and a response to this comment by Ahmadi et al., has nuanced this result. While Remya et al. conclusions are true in the linear regime, Ahmadi et al. showed that, in the non-linear evolution, the anisotropic electrons will be unstable to the electron whistler instability. This instability, in absence of heavy ions, will lower the anisotropy level and thus will restore the previous balance in which the proton cyclotron instability dominates over the mirror instability. We have added this discussion to the revised manuscript lines 255 - 268 as a warning to the readership. Future extensions of the hybrid formalism may investigate the role electrons have, but no such models exist at this time. We believe that our results are valid in the context of the hybrid approach and are of interest to future global-hybrid modelling efforts. We have added that, despite the importance of this discussion, our conclusion remains that at the lowest spatial resolution, the range of k-vectors the simulation can model is not large enough to allow the proton cyclotron instability to develop, whether it dominates over the mirror instability or not.**
**We have also added a reference to Soucek et al. 2015 showing that, for Mach number $< 7$, $M_A = 6.9$ in our study, the proton cyclotron should dominate in the magnetosheath. Thus, the investigation of how cyclotron instabilities can be simulated is important.**

Also according to several of your references, the code must include other heavier ions such as helium and oxygen to generate stop bands and lower the growth rate of ion cyclotron waves. Has

this been done for these simulations (I see later in the paper that the answer is "no")?

**We thank the reviewer for pointing out that we have not specified that it has not been done in our simulations. We have made it more clear in the revised manuscript, line 271.**

Your simulation results indicate that in some cases you will have growth of ion cyclotron waves from the bow shock to midway through the magnetosheath and mirror waves closer to the magnetopause. I believe that such a case has never been seen before. You might wish to check the many references on mirror mode waves in the 2011 JGR review paper (as suggested before). Mirror modes are typically observed throughout the entire magnetosheath. See examples in the 1982 paper for the magnetosheaths of the Earth, Jupiter and Saturn. Proton cycloton waves are rarely detected in the Earth's magnetosheath. This was what started the whole mirror mode/proton cyclotron growth rate debate. Early theoretical work (JGR 97, 8519, 1992; JGR 98, 1481–1488, 1993) predicted that the sheath would be filled with proton cyclotron waves whereas observations only detected mirror mode waves. Therefore these early works put in unrealistic Helium densities to lower the proton cyclotron wave growth rate (the average solar wind density is only 4%). [...]
When proton cyclotron waves have been detected, they have been observed throughout the entire magnetosheath. See Proc. COSPAR Coll., edited by L.-H. Lyu, Pergamon Press, 97, 2000 (mentioned before). This observation should be mentioned and discussed. I believe your current "predictions" are contrary to observations and will be misleading to the readership.

**We thank the reviewer for highlighting the lack of clarity of one of our statement. We have added clarifications line 275-277 to highlight that instabilities grow near the bow-shock, and then the resulting waves are transported throughout the magnetosheath, where they are observed. We have added the several references regarding detection of the proton cyclotron waves in the Earth's magnetosheath (lines 44-48 in the introduction) and their dominance under conditions similar to the ones in our simulations.**

Although you have indicated the main intent of your paper is to illustrate the usefulness of your code, by using mirror modes and proton cyclotron waves as your examples, you cannot escape the issue of why mirror modes dominate.

**We agree that the physics included in the simulation model will affect which modes dominate the sheath, but note that there is evidence that our physics model is adequate for this problem. This investigation asserts that certain resolutions however prevent the ion cyclotron instability from triggering, causing the growth of proton anisotropy and further divergence for the results from the value expected from the physical model. It might appear as the mirror mode waves/structures dominate even in the high-resolution run in Figure 1a of the manuscript which shows magnitude of B. Since the proton cyclotron waves are only slightly compressive, they do not clearly show up in this figure. When instead plotting the out-of-plane magnetic field (By), the proton cyclotron waves appear much more clearly in the 300 km case, see attached figure. Moreover, Fig. 2a and 3a show that the proton cyclotron waves around the cyclotron frequency are the most prominent feature in our simulations. Therefore**

[Figure]

[Figure]

[Figure]

[Figure]

Figure 1: Out-of-plane component ($B_y$) of the magnetic field at the highest (a) and lowest (c) resolution.

**we conclude that both instabilities are present in our simulation, as predicted, and that the proton cyclotron instability dominates in the simulation at higher resolution, while it cannot develop at the lowest resolution. We also added, lines 266-268, as stated above, that "at the lowest resolution, the range of k-vector the simulation is able to model is not large enough to allow the proton cyclotron instability to grow, regardless of which mode is expected to dominate" which is our main conclusion, since the goal of this paper is to determine the effect of spatial resolution on these instabilities.**

The Remya et al 2014 paper shows that the proton cyclotron waves in the magnetosheath are elliptically polarized (not circular) and are propagating at large angles (not parallel) to the ambient magnetic field. Can your simulations predict that? If not, you need to explain to the readership.
Line 29, "left handed circularly polarized wave". As mentioned previously the waves have been observed to be elliptically polarized.
Line 166, "theta kB is 15 deg". As mentioned previously, this is not what has been observed experimentally.

We searched the literature, but the best match for a paper discussing the topic we could find was **DOI:10.1088/0004-637X/793/1/6**. This paper states that "Thus, almost all waves studied are consistent with their being electromagnetic proton cyclotron waves. Most of the waves ($\approx 55\%$) were found to be propagating along $B_0$ ($\theta_{kB_0} < 30°$), as expected from theory. However, a significant fraction of the waves were found to be propagating oblique to $B_0$. These waves were also circularly polarized.", which is consistent with our results. Our hodograms in Figure 4 also show some degree of ellipticity to the polarization. Thus, we have changed the text to reference nearly circular polarization. We have also cited this paper for observations of the proton cyclotron instability in the Earth's magnetosheath.**

Introduction, line 21. The standard reference for the proton cyclotron instability is JGR, 71, 1-28, 1966. I suggest adding this here.

**We thank the reviewer for providing this reference. We have added it to the revised manuscript.**

Line 26, "magnetic perturbations which are parallel to the background magnetic field". I suggest adding the Tsurutani et al. 1982 reference which was the first to show it experimentally and Price et al. 1986 who were the first to show it via simulations.

**We thank the reviewer for providing these references. We have added them to the revised manuscript.**

Line 37: "cometary sheaths". I suggest adding the references: 98, 20,955–20,964, 1993 and NPG, 6, 229–234, 1999. doi:10.5194/npg-6-229-1999.

**We thank the reviewer for providing these references. We have added them to the revised manuscript.**

Line 59, "Kinetic Alfven Wave Turbulence". Please give references here. The authors should note that not everyone agrees what this is. For example in JGRSP, 123, 2018, https://doi.org/10.1002/2017JA024203 the authors have shown that nonlinear Alfven wave turbulence is nothing like what is predicted by (linear) theory.

**We have added references on this topic to the revised manuscript.**

Lines 244-252. Here you mention that the addition of electron anisotropy and cold helium ions will change your results. This should be emphasize more and your conclusions about the usefulness of this code should be tempered a bit. Real plasmas can be more complex.
Lines 260-261 and the McKean et al. 1992 reference. This is a simulation result and clearly does not match observations very well (as previously mentioned). I suggest tempering your remarks relative to this paper here.

**As mentioned before, we have added a discussion on this topic lines 255-268. We have also added clarity to the later statement, lines 271-272.**

**Relevant changes made to manuscript**

**Introduction**

- Added references provided by the reviewers.

- Added a few sentences about observations of the proton cyclotron waves in the Earth's magnetosheath.

**Discussion**

- Added a paragraph discussing the effect of the anisotropic electron distributions on the study.

- Clarified a few sentences regarding the location of waves in the simulation.

**Tracked changes Manuscript**

Following is the revised manuscript. The discussions and minor revisions added for clarity can be found in bold font. The sentences removed from the manuscript can be found crossed.

[revised manuscript text omitted]

---

## Author Response (AR4)

**Response to reviewer #1**

**We thank the reviewer for their examination of our manuscript and their constructive comments. Please find below our response to your concerns, in bold font. The lines indicated in red in our response are pointing to the lines in the revised manuscript with tracked changes. It can be found at the end of this document after the list of relevant changes. The lines in blue in our response are pointing to the lines in the revised manuscript, which has been resubmitted in a separate file.**

If the authors wish to include comments from Ahmadi et al. (2016), they should also include the Comment by Remya et al. (2017) and Reply by Ahmadi et al. (2017). The latter two are: JGRSP, 122, 745-747, doi:10.1002/2016JA023148, 2017 and JGRSP, 122, 748-752, doi:10.1002/2016JA023452, 2017, respectively.

**We have added these references in the discussion on the matter (Lines 259-262 in the revised manuscript with tracked changes and Lines 256-259 in the revised manuscript.)**

**Following is a summary of our previous response, with the corresponding lines of the changes previously made in the revised manuscript with tracked changes and revised manuscript.**

- Regarding the anisotropic electron distributions: **We have added this paragraph in the discussion:** "Remya et al. (2013) showed that an anisotropic electron distribution, which is not modeled in Vlasiator, with $T_{\perp,e}/T_{\parallel,e} > 1.2$ reduces the proton cyclotron instability growth rate, while increasing the mirror instability growth rate. In this case, the mirror instability will dominate over the proton cyclotron instability. Masood and Schwartz (2008) showed that the Earth's magnetosheath presents such anisotropic electron distributions. A later study by Ahmadi et al. (2016), a comment by Remya et al. (2017) and a reply to this comment by Ahmadi et al. (2017) nuance this result. While Remya et al. (2013) conclusions hold true in the linear regime, Ahmadi et al. (2016) showed that, in the non-linear evolution, the anisotropic electron distributions will be unstable to the electron whistler instability. This instability, in the absence of heavy ions, will lower the anisotropy level and thus will restore the previous balance between the proton cyclotron instability and the mirror instability. To this day, there are no global-hybrid Vlasov simulations able to model both protons and anisotropic electrons in the Earth's magnetosheath to confirm these results. Fig. 5 of Soucek et al. (2015) also showed that the proton cyclotron waves should dominate in the magnetosheath for Mach numbers lower than 7. The Mach number in our simulation is

$M_A = 6.9$. With all these factors considered, we expect both instabilities to be present in our simulations, and the proton cyclotron instability to have a higher maximum growth rate than the mirror instability in our simulations, when both modes are resolved. However, at the lowest resolution, the range of k-vectors the simulation is able to model is not large enough to allow the proton cyclotron instability to grow, regardless of which mode is expected to dominate." (**Lines 255-270 and Lines 253-267**).

- Regarding the absence of heavy ions in our simulations: **We specified it with a sentence (Lines 272-273 and Lines 269-270).**

- Regarding the location of both instabilities in our simulations: **We clarified our statement:** "Both kinetic instabilities grow near the quasi-perpendicular bow shock, where the temperature anisotropy is higher, and the resulting waves and structures propagate and travel with the plasma flow in the magnetosheath." (**Lines 277-279 and Lines 273-275**). **We have also added references of their observation in the magnetosheath and the dominance of the proton cyclotron waves under conditions similar to our simulations in the introduction:** "Although the Earth's magnetosheath tends to have a high plasma beta, there are also many reports of observations of proton cyclotron waves (Remya et al., 2014; Soucek et al., 2015; Zhao et al., 2018, 2020). Soucek et al. (2015) show observations in the Earth's magnetosheath of proton cyclotron waves associated with Alfvén Mach number below 7, which is within the typical Mach number range at Earth, between 6 and 8 (Winterhalter and Kivelson, 1988)". (**Lines 44-48 and Lines 44-47**).

- Regarding the "issue of why mirror modes dominate": **The discussion and figure included in our response was meant to convince the reviewer that both instabilities are present in our simulations and that the proton cyclotron instability dominates, as predicted. This was already stated in our manuscript, therefore nothing was added regarding this discussion (besides the comments previously discussed). We have added the statement:** "However, at the lowest resolution, the range of k-vectors the simulation is able to model is not large enough to allow the proton cyclotron instability to grow, regardless of which mode is expected to dominate." (**Lines 269-270 and Lines 265-267**). **We have also emphasised this in our conclusion (Line 366 and Line 352).**

- Regarding the ellipticity and propagation angle of the proton cyclotron instability: **We have added the referenced paper in the introduction (Lines 30 and 45 in both versions of the manuscript). We have also changed every mention of "circularly polarised" to "nearly circularly polarised" (Lines 29, 235, 327, 343 and Lines 29, 234, 323, 339). We apologise for not highlighting this change in bold in the revised manuscript with tracked changes previously, this is now the case.**

- **We have added the mentioned references: regarding the proton cyclotron instability (Line 21 in both version of the manuscript, not in bold in the revised manuscript with tracked changes due to LaTeX formatting), regarding the mirror instability (Line 26 in both versions of the manuscript), cometary sheaths (Lines 38-39 in both versions of the manuscript, not in bold in the revised manuscript with tracked changes due to LaTeX formatting), Kinetic Alfven Waves (Line 64 and Line 63).**

**Additionally, we have modified the paragraph regarding McKean et al. (1994) (previously Lines 266-269 in the revised manuscript) as such:** "McKean et al. (1994) showed that introducing Helium ions in a simulation tend to suppress the proton cyclotron instability (supported by Remya et al. (2013)), with only the mirror instability remaining. While there are no heavy ions in our study, our results display similar wave properties in the magnetosheath with a resolution of 900 km, when the proton cyclotron instability does not develop." (**Lines 269-274 and Lines 268-271**).

**We have also added a sentence in the conclusion (Lines 366-367 and Lines 352-353):** "The predominance of wave modes such as the proton cyclotron and mirror modes in the magnetosheath is an active topic of research.". **We emphasised that our study** "shows that the proton cyclotron instability does not develop at low spatial resolution, assuming the plasma conditions allow it to develop in the first place." **in our conclusion (Lines 368-369 and Lines 354-355).**

**Relevant changes made to manuscript**

**Introduction**

- Added references provided by the reviewers.

- Added a few sentences about observations of the proton cyclotron waves in the Earth's magnetosheath.

**Discussion**

- Added a paragraph discussing the effect of the anisotropic electron distributions on the study. Added references provided by the reviewer.

- Clarified a few sentences regarding the location of waves in the simulation.

- Modified a paragraph regarding heavy ions in simulations.

**Conclusions**

- Added a sentence regarding the state of the current work.

- Added emphasis on the conditions when the study is valid.

**Other changes**

- Modified "circularly polarised" to "nearly circularly polarised" throughout the manuscript.

**Tracked changes Manuscript**

Following is the revised manuscript. The discussions and minor revisions added for clarity can be found in bold font. The sentences removed from the manuscript can be found crossed.

[revised manuscript text omitted]

---

## Author Response (AR5)

**Response to reviewer #1**

We thank the reviewer for the examination of our manuscript and their comments.

We respectfully disagree with their view on simulation codes. The Referee states that there are two purposes for simulations, 1) predicting observational results and 2) explaining observations that have been made. We agree with this categorisation, but maintain that before we can do 1) - 2), the 3) codes need to be numerically validated and tested. A significant part of modelling papers, basically all papers dealing with preparing space weather predictions, have to do with this point 3), see e.g., the papers we list below. We also maintain that this validation and testing must be openly refereed according to rigorous scientific principles. Failing to do so casts a doubt to the codes - how can users of the codes trust the modelling results if only the results are published? If the validation and testing is not published according to rigorous scientific principles, it leads to a situation where some factors and parameters of the codes are hidden, and the choices that the modellers make may not be based on scientific principles but e.g., on personal feelings.
We therefore expect critical and fair feedback that treats our work on par with comparable studies published previously in Annales Geophysicae. We all have the same goal to understand and explain observational results while nonetheless being aware that a collisionless plasma spanning several orders of magnitudes in spatial and temporal scales can not simply be tackled self-consistently with today's computational capacities. It also would be unrealistic, and in our opinion unfair, to require of a global code like ours to perform as well as a local one (and vice-versa, since what local codes gain in spatial and temporal accuracy prevents them from accounting for multiple boundaries interactions and large-scale inhomogeneities). Our current study is not the final word on the topic, but an early incremental and important one, that will hopefully serve as a springboard for future validation studies we and others will pursue.

We also believe that we have provided sufficient references to show that our simulations match the observations of the proton cyclotron and mirror instabilities in the magnetosheath. We would like to point out that Hoilijoki et al. (2016) have already conducted a study regarding the simulation of mirror modes in the magnetosheath using the global-hybrid Vlasov simulation Vlasiator, and that they are similar to observations.

Regarding the location of both modes in the magnetosheath, we have modified Fig.

**7 to add the figure sent in our previous response, and added the following discussion to the manuscript:** "Panels (a)-(c) of Fig. 7 display the y-component (out of plane) of the magnetic field at the three different resolutions. The AIC waves are clearly present throughout the magnetosheath at the highest resolution, and have completely disappeared at the lowest resolution, as previously predicted by Fig. 6. The intermediate resolution displays that the AIC waves are still present in the simulation, but at lower amplitude." **(Lines 290-293 in both the revised manuscript and the revised manuscript with tracked changes, and the corresponding paragraph in the Results section, lines 201-204.)**

**Relevant changes made to manuscript**

**Figures**

- Modified and added three panels to Figure 7.

**Results**

- Added a paragraph related to the addition of new panels in Figure 7.

**Discussion**

- Added a paragraph related to the addition of new panels in Figure 7.

**Tracked changes Manuscript**

Following is the revised manuscript. The discussions and minor revisions added for clarity can be found in bold font. The sentences removed from the manuscript can be found crossed.

[revised manuscript text omitted]